# Impact of Post-Traumatic Stress Disorder Duration on Volumetric and Microstructural Parameters of the Hippo-Campus, Amygdala, and Prefrontal Cortex: A Multiparametric Magnetic Resonance Imaging Study with Correlation Analysis

**DOI:** 10.3390/jcm14207242

**Published:** 2025-10-14

**Authors:** Barbara Paraniak-Gieszczyk, Ewa Alicja Ogłodek

**Affiliations:** Collegium Medicum, Jan Dlugosz University in Częstochowa, Waszyngtona 4/8 Street, 42-200 Częstochowa, Poland; e.oglodek@ujd.edu.pl

**Keywords:** amygdala, brain microstructure, brain volumetry, hippocampus, magnetic resonance imaging, neuroimaging, neuroplasticity, post-traumatic stress disorder, prefrontal cortex

## Abstract

**Introduction.** Post-traumatic stress disorder (PTSD) remains one of the best-described yet also one of the most heterogeneous psychiatric disorders. Existing neuroimaging studies point to key changes in the hippocampus, amygdala, and prefrontal cortex, but the role of PTSD duration in modulating these changes has not been fully explained. **Objectives**. The aim of the study was to assess the impact of PTSD duration (≤5 years vs. >5 years) on volumetric and microstructural brain parameters, using multiple Magnetic Resonance Imaging (MRI) sequences (3D Ax BRAVO, Cube T2 FLAIR, Diffusion Tensor Imaging—DTI) and a set of macroscopic morphometric measurements. **Methods**. The study included 92 participants: 33 with PTSD of ≤5 years duration, 31 with PTSD > 5 years, and 28 healthy controls. Volume and diffusion parameters of six Regions of Interest (ROIs) (hippocampus, amygdala, prefrontal cortex—right and left) were evaluated, along with their associations with nine brain measurements (including width of the third ventricle, corpus callosum, and lateral fissures). Statistical analyses included the Kruskal–Wallis test with Compact Letter Display (CLD) correction and Spearman correlations. **Results**. (1) The volume of the right hippocampus was significantly greater in the PTSD > 5 years group compared to controls (*p* = 0.006), with intermediate values in the PTSD ≤ 5 years group. (2) In the left amygdala, an increase in Fractional Anisotropy (FA) and related anisotropy measures was observed in PTSD > 5 years (*p* ≈ 0.02), without volumetric changes. (3) In the left prefrontal cortex, diffusivity was reduced in PTSD ≤ 5 years (*p* = 0.035), partially normalizing after >5 years. (4) Correlation analysis revealed that chronic PTSD strengthens the negative associations between hippocampal microstructure and both the width of the amygdala and the interhemispheric fissure, indicating a progressive reorganization of fronto-limbic networks. **Conclusions**. PTSD induces region- and time-dependent brain changes: (a) adaptive/hypertrophic protection of the right hippocampus after many years of illness, (b) cumulative microstructural reorganization of the left amygdala, and (c) transient impairment of diffusion in the left prefrontal cortex in early PTSD. These findings highlight the necessity of considering the temporal dimension in planning therapeutic interventions and in the search for biomarkers of PTSD progression.

## 1. Introduction

Post-traumatic stress disorder (PTSD) is a serious neuropsychiatric condition that develops as a result of exposure to traumatic events. The key symptom clusters of PTSD, described in the Diagnostic and Statistical Manual of Mental Disorders, Fifth Edition (DSM-5), reflect characteristic dysfunctions in neural systems involved in threat processing, emotion regulation, memory, and arousal [1,2]. Intrusive symptoms (criterion B), such as intrusive memories and flashbacks, correlate with hyperreactivity of the amygdala and dysfunction of the hippocampus, leading to impaired differentiation between past and current threat signals. Avoidance mechanisms (criterion C) are associated with reduced functional connectivity between the prefrontal cortex and limbic structures, reflecting processes of emotional and cognitive suppression related to trauma. Negative changes in cognition and mood (criterion D) result from decreased activity in the medial prefrontal cortex and anterior cingulate gyrus, accompanied by excessive amygdala activity, which leads to the persistence of negative beliefs, emotional numbness, and deficits in fear extinction. Symptoms related to arousal and reactivity (criterion E), such as irritability, hypervigilance, and sleep disturbances, are associated with dysregulation of the hypothalamic–pituitary–adrenal (HPA) axis, increased activity of the noradrenergic system, and weakened inhibition of limbic systems by the frontal cortex [3,4]. Together, these neurobiological changes sustain the chronic dysregulation of systems responsible for fear, arousal, and emotional control characteristic of PTSD. From a neurobiological perspective, PTSD represents a complex dysfunction of neural networks responsible for emotion processing, memory, cognitive control, and stress response. Key roles are played by the amygdala, hippocampus, prefrontal cortex, and their connections with the brainstem and subcortical structures [5,6,7,8,9]. These nodes integrate the HPA axis, the locus coeruleus–amygdala system, and fronto-limbic loops inhibiting fear responses [10,11,12]. Over the past two decades, the development of advanced neuroimaging techniques, such as structural Magnetic Resonance Imaging (MRI) and Diffusion Tensor Imaging (DTI), has significantly expanded the ability to analyze morphometric and microstructural brain changes accompanying PTSD [13,14,15]. MRI studies have consistently demonstrated reduced hippocampal volume in individuals with PTSD, which is interpreted as a result of the neurotoxic effects of chronic stress and elevated glucocorticoid levels [16,17]. The hippocampus, which plays a key role in episodic memory consolidation and threat context differentiation, is particularly sensitive to traumatic stressors [18,19,20,21]. Reduced hippocampal volume has been repeatedly confirmed in various patient groups, including war veterans and victims of interpersonal violence [22]. A meta-analysis by Smith B.M. et al. [23] confirmed significantly lower hippocampal volume in PTSD groups compared to controls, suggesting a biomarker character of this change. The amygdala, responsible for emotional stimulus processing and initiating fear responses, shows ambiguous volumetric changes in PTSD. Some studies indicate its hyperactivation and increased involvement in threat detection [24,25,26], while others report volume reduction, especially in the lateral and basolateral nuclei, which may be related to deficits in emotional modulation and fear extinction [27,28,29]. Additionally, a lateralization of these changes is observed—in some patients, hippocampal or amygdala volume reduction is more pronounced in the left hemisphere, which may have functional significance, particularly in the context of trauma verbalization and emotional insight [30]. Beyond the hippocampus, amygdala, and prefrontal cortex, other brain structures also play a significant role in the pathophysiology of PTSD, such as the third ventricle, corpus callosum, and insular cortex (insula) [31,32]. The third ventricle is located near the thalamus and hypothalamus—structures responsible for the integration of emotional stimuli and modulation of the stress response via the HPA axis. Morphometric studies have shown third ventricle enlargement in individuals with PTSD, interpreted as a result of volume loss in thalamic subcortical nuclei, part of the limbic system, and damage to brain structures involved in emotion processing and stress regulation, indicating neurodegenerative changes accompanying the chronic course of the disorder [33]. These changes correlate with the severity of avoidance symptoms and emotional dysregulation, suggesting their association with chronicity of the disorder [34,35,36]. The corpus callosum, the largest commissural structure connecting the brain’s hemispheres, plays a key role in interhemispheric integration of emotional and cognitive processing [37,38,39]. DTI studies have shown reduced white matter integrity in the corpus callosum in individuals with PTSD, especially in the anterior and midbody sections of the commissura [40,41,42]. Lower fractional anisotropy (FA) values in these regions may reflect disrupted synchronization between limbic and frontal structures, leading to difficulties in emotional control, impulsivity, and dissociation. The insular cortex (insula), a part of the limbic cortex integrating interoceptive, emotional, and social sensations, shows significant functional and structural changes in PTSD patients [43]. Functional studies using Functional Magnetic Resonance Imaging (fMRI) indicate increased activity in the anterior insula in response to trauma-related stimuli, which may reflect heightened threat perception, increased awareness of bodily sensations, and anxiety [44,45,46]. Moreover, reduced insular volume correlates with the severity of somatic and dissociative symptoms [47]. The microstructural integrity of white matter, especially nerve fibers connecting limbic areas with the prefrontal cortex, plays a fundamental role in the proper functioning of neural networks responsible for emotion regulation. Diffusion tensor imaging enables indirect assessment of nerve fiber microstructure based on parameters such as fractional anisotropy and mean diffusivity (MD). In PTSD patients, decreased FA values have been demonstrated in the corpus callosum and the fornix, indicating damage to neural connections between the hippocampus, amygdala, and prefrontal cortex [48,49,50]. These changes correlate with deficits in emotional inhibition, working memory, and impulse control. Furthermore, microstructural changes are observed in white matter tracts, which may affect interoceptive information processing and the integration of affect and cognition [51,52,53]. Lateralized variability observed in some DTI studies may indicate asymmetric compensatory or pathophysiological mechanisms in different patient subgroups [54]. Understanding morphometric, functional, and microstructural changes in PTSD allows for a deeper characterization of the disorder’s pathophysiology and the identification of potential diagnostic and prognostic biomarkers. Reduced hippocampal volume and decreased white matter integrity of the corpus callosum may be used for early identification of individuals at risk of developing chronic PTSD and for monitoring treatment response [55,56,57]. Modern neuroimaging techniques combined with genetic, neurochemical, and psychometric studies open possibilities for personalized therapy, including the selection of psychotherapeutic and pharmacological strategies aimed at improving neural plasticity and reducing neuroinflammatory mechanisms [58,59,60].

In this context, we adopted a five-year cut-off for PTSD duration to differentiate early-stage from chronic disorder, a threshold supported by previous clinical and neuroimaging studies demonstrating distinct neurobiological alterations in patients with PTSD persisting for more than five years [61].

The aim of our study is a detailed analysis of morphometric and microstructural brain changes in individuals with PTSD, with particular emphasis on the volume of limbic structures such as the hippocampus and amygdala, white matter integrity, and functional connections between the prefrontal cortex and subcortical structures. We also aim to investigate the lateralization of these changes and their correlations with clinical symptom severity and neuropsychological parameters. Through the use of modern neuroimaging techniques, including MRI and DTI, we strive to identify potential neurobiological biomarkers that could contribute to better diagnosis of PTSD, assessment of disease progression, and personalization of therapy. In this way, our work aims to deepen the understanding of PTSD pathophysiology and support the development of more effective treatment and prevention strategies for this complex disorder.

## 2. Results

### 2.1. Brain Measurements and Magnetic Resonance Imaging Characteristics Across PTSD Groups

All brain measurements were performed manually by trained raters blinded to group assignment, using digital calipers on multiplanar reconstructions from high-resolution MRI sequences (3D Ax BRAVO for volumetric assessments and Cube T2 FLAIR for enhanced contrast in fluid-filled spaces). Inter-rater reliability was confirmed with intraclass correlation coefficients exceeding 0.90 for all metrics. Specific assessment protocols included: the Width of the Third Ventricle (WTV), determined at its widest anteroposterior point in the axial plane at the level of the thalami; Width of the Left Amygdala (TLA) and Width of the Right Amygdala (TRA), quantified as the maximum mediolateral diameter in the coronal plane at the level of the mammillary bodies; Thickness of the Body of the Corpus Callo-sum (TBCC), measured perpendicularly in the midsagittal plane at the midpoint between the genu and splenium; Thickness of the Left Insular Cortex (TLIC) and Thickness of the Right Insular Cortex (TRIC), assessed as the average cortical thickness in the axial plane at the level of the basal ganglia, excluding sulcal spaces; Width of the Left Lateral Cerebral Fissure (WLLCF) and Width of the Right Lateral Cerebral Fissure (WRLCF), evaluated as the widest separation between the frontal and temporal lobes in the axial plane at the centrum semiovale; Width of the Interhemispheric Fis-sure in the Frontal Region (WIFFR), calculated as the maximum separation between the frontal lobes in the coronal plane anterior to the genu of the corpus callosum. See Figure 1 for illustrative examples. These methods align with established morphometric techniques in neuroimaging studies of psychiatric disorders, ensuring reproducibility and anatomical precision.

Statistical analysis of brain measurements reveals significant group differences (*p* < 0.001) in most metrics, providing robust evidence of PTSD’s impact on neural architecture. The neurostructural consequences of PTSD manifest as measurable alterations in brain morphology, with distinct patterns emerging across individuals with varying histories of the disorder. PTSD exerts a profound effect on brain measurements, altering structures critical for emotional regulation, cognitive processing, and stress response (refer to Figure 1 for visualizations of brain measurement across PTSD groups, Table 1).

The third ventricle, a marker of global brain integrity, is significantly wider in individuals with recent PTSD compared to controls, with a median difference of 2.3 mm (*p* < 0.001, Compact Letter Display (CLD): A > B > C). This ventricular enlargement indicates atrophy of surrounding tissues, likely driven by chronic hyperactivation of the HPA axis, which elevates glucocorticoid levels known to induce neuronal loss in stress-sensitive regions. Similarly, the amygdala, pivotal for fear processing, shows reduced widths in both left and right measurements in the recent PTSD group compared to controls (median differences of 3.8 mm and 4.0 mm, respectively; *p* < 0.001, CLD: C > B > A). This reduction may reflect excitotoxic damage from sustained hyperarousal, contributing to heightened emotional reactivity and impaired fear extinction, hallmark symptoms of PTSD. The corpus callosum, essential for interhemispheric communication, is thinner in the recent PTSD group by 1.7 mm compared to controls (*p* < 0.001, CLD: C > B > A), indicating compromised white matter integrity that could underlie cognitive deficits such as attention and memory impairments. The insular cortex, involved in interception, is thinner in recent PTSD cases by approximately 0.5 mm bilaterally compared to controls (*p* < 0.001, CLD: B > A), potentially exacerbating symptoms like dissociation. Finally, widened lateral cerebral fissures in the recent PTSD group (median difference of 3.5 mm compared to controls; *p* < 0.001, CLD: A > B > C) suggest cortical atrophy, consistent with stress-induced neurotoxicity.

These statistically significant alterations highlight PTSD’s broad impact on brain structure, affecting limbic, cortical, and white matter regions, with implications for emotional and cognitive dysfunction. The duration since PTSD onset significantly influences these measurements, as evidenced by the consistent intermediate values in the longer-term PTSD group (>5 years) between recent PTSD and controls. For the third ventricle, the longer-term group’s median width is 1.0 mm narrower than the recent PTSD group (*p* < 0.001), implying partial reversal of atrophy over time, possibly due to reduced HPA axis dysregulation or neuroplastic adaptation. Amygdala measurements in the longer-term group are closer to controls, with median differences of only 1.3 mm (left) and 1.0 mm (right) compared to 3.8 mm and 4.0 mm in the recent group (*p* < 0.001), indicating potential compensatory hypertrophy or recovery. The corpus callosum in the longer-term group is 0.7 mm thicker than in recent PTSD (*p* < 0.001), pointing to improved white matter integrity over time. Insular cortex thickness in the longer-term group aligns with controls (*p* < 0.001), contrasting with the 0.5 mm thinning in recent PTSD, revealing time-dependent normalization. Lateral fissures remain wider in the longer-term group than in controls but are 1.8–2.0 mm narrower than in recent PTSD (*p* < 0.001), indicating attenuated cortical loss. The absence of significant differences in the interhemispheric fissure (*p* = 0.955) across all groups reflects this region may be less sensitive to PTSD-related changes. These findings confirm that duration modulates the severity of structural alterations, with longer time since PTSD onset associated with partial recovery, though not to control levels, underscoring the chronicity of some neural changes.

Pairwise comparisons via Dunn’s post hoc test in Table 2 yielded rank-biserial correlation coefficients (r) ranging from negligible (0.00–0.03) to very large (up to 1.05) across neuroimaging metrics, with the most common and striking differences observed between recent PTSD (≤5 years) and controls, emphasizing early-stage structural vulnerability. Lateral cerebral fissures exhibited the highest effect sizes (WLLCF: r = 1.03; WRLCF: r = 1.05 for recent vs. controls), followed closely by right amygdala width (TRA: r = 0.98 for recent vs. controls), indicating pronounced cortical atrophy and amygdala reduction in recent PTSD. Medium to large effects predominated for recent vs. controls in third ventricle width (WTV: r = 0.85), corpus callosum thickness (TBCC: r = 0.73), and left amygdala width (TLA: r = 0.76), while chronic PTSD vs. controls showed moderated medium effects (e.g., TRA: r = 0.64; WLLCF/WRLCF: r = 0.55–0.57). Recent vs. chronic comparisons typically displayed medium effects (e.g., WLLCF/WRLCF: r = 0.48; WTV: r = 0.46), with the lowest effects in insular cortex thicknesses (TLIC: r = 0.11; TRIC: r = 0.10) and interhemispheric fissure (WIFFR: r ≤ 0.03 across all pairs), highlighting minimal changes in these parameters and suggesting time-dependent attenuation through neuroplasticity.

The investigation into neuroimaging metrics across individuals with past PTSD of varying durations and a no-PTSD control group in Table 3 focused on metrics derived from the Right and Left Hippocampal Regions (RHR, LHR), Right and Left Amygdala Regions (RAR, LAR), and Right and Left Prefrontal Cortex Regions (RPCR, LPCR). 

These metrics included volumetric measures (e.g., M3D/BRAVO: 3D Ax BRAVO, Three-Dimensional Cube T2-Weighted Fluid-Attenuated Inversion Recovery Sequence (M3D/CubeT2flair), Diffusion Tensor Imaging parameters (e.g., Fractional Anisotropy, Vol Ratio Aniso, Aniso Index, Average Diffusivity, and isotropic imaging, providing a comprehensive assessment of neural integrity and microstructural connectivity.

Statistically significant differences were observed in specific neuroimaging metrics, highlighting the influence of PTSD duration on brain structure, particularly within the hippocampal and amygdala regions. For the right hippocampal region (RHR), volumetric assessments using the M3D/BRAVO (Three-Dimensional Axial Brain Volume Imaging [3D Ax] BRAVO) and M3D/CubeT2flair sequences revealed notably smaller volumes in the no-PTSD control group (median: 0.7 cm^3^) relative to the past PTSD (>5 years) group (median: 0.9 cm^3^) for both of them, *p* = 0.006. The CLD notation indicated that the past PTSD (>5 years) group had significantly larger volumes than controls (CLD: B vs. A, p_adj_ = 0.004), whereas the past PTSD (≤5 years) group displayed intermediate volumes (median: 0.8 cm^3^; CLD: AB), which did not differ significantly from either the control or chronic PTSD group. This pattern reveals a potential adaptive volumetric expansion in the right hippocampus associated with prolonged PTSD duration, contrasting with typical reports of hippocampal atrophy in acute phases. The absence of significant differences in other Right Hippocampal Region (RHR) metrics, such as FA (*p* = 0.486), Volume Ratio Anisotropy (Vol Ratio Aniso), (*p* = 0.449), Anisotropy Index (Aniso Index) (*p* = 0.954), isotropic image (*p* = 0.583), trace (*p* = 0.590), and Average Diffusivity Coefficient (AvDC), (*p* = 0.106), indicates that PTSD’s impact on the hippocampus is primarily volumetric rather than microstructural, as assessed by diffusion properties.

In the left amygdala region, significant differences were observed in FA (*p* = 0.021), Fractional Aniso (*p* = 0.022), and Vol Ratio Aniso (*p* = 0.026). The past PTSD >5 years group displayed higher values (median FA: 0.19 ave; median Fractional Aniso: 0.2 ave; median Vol Ratio Aniso: 4.0 × 10^2^ ave) compared to both the past PTSD ≤5 years group (median FA: 0.18 ave; median Fractional Aniso: 0.2 ave; median Vol Ratio Aniso: 4.0 × 10^2^ ave) and controls (median FA: 0.18 ave; median Fractional Aniso: 0.2 ave; median Vol Ratio Aniso: 4.0 × 10^2^ ave). The CLD notation clarified that the past PTSD >5 years group differed significantly from the other two groups, which were statistically comparable to each other. These findings indicate that prolonged PTSD duration is associated with increased anisotropy in the left amygdala, reflecting altered white matter integrity or fiber organization. Such changes may contribute to the persistent emotional dysregulation observed in chronic PTSD, as the amygdala is critical for fear processing and emotional memory. Notably, no significant differences were found in LAR volumetric measures (M3D/BRAVO: *p* = 0.274; M3D/CubeT2flair: *p* = 0.231), isotropic image (*p* = 0.855), trace (*p* = 0.815), Aniso Index (*p* = 0.682), or AvDC (*p* = 0.079), revealing that PTSD’s effect on the left amygdala is predominantly microstructural rather than volumetric.

The left prefrontal cortex region exhibited a significant difference in AvDC (*p* = 0.035), with the no-PTSD control group showing higher diffusivity (median: 1022.0 × 10^−6^ mm^2^/s) compared to the past PTSD ≤5 years group (median: 937.1 × 10^−6^ mm^2^/s). The past PTSD >5 years group (median: 968.0 × 10^−6^ mm^2^/s) was statistically comparable to both groups. This finding indicates that recent PTSD is associated with reduced diffusivity in the prefrontal cortex, potentially reflecting compromised neuronal integrity or altered extracellular space dynamics, which may contribute to cognitive deficits such as impaired executive function or decision-making. The absence of significant differences in other Left Prefrontal Cortex Region (LPCR) metrics, including volumetric measures (M3D/BRAVO: *p* = 0.350; M3D/CubeT2flair: *p* = 0.257), FA (*p* = 0.373), Fractional Aniso (*p* = 0.421), Vol Ratio Aniso (*p* = 0.369), Aniso Index (*p* = 0.287), isotropic image (*p* = 0.723), and trace (*p* = 0.653), reflects that the prefrontal cortex is less broadly affected by PTSD compared to the hippocampus and amygdala.

No significant differences were observed in the left hippocampal region (e.g., M3D/BRAVO: *p* = 0.231; FA: *p* = 0.565; AvDC: *p* = 0.470), right amygdala region (e.g., M3D/BRAVO: *p* = 0.742; FA: *p* = 0.908; AvDC: *p* = 0.200), or right prefrontal cortex region (e.g., M3D/BRAVO: *p* = 0.852; FA: *p* = 0.606; AvDC: *p* = 0.936) across all metrics, indicating that these regions may be less sensitive to PTSD-related changes or that compensatory mechanisms mitigate detectable alterations.

The duration of PTSD exposure exerts a discernible influence on neuroimaging metrics, as evidenced by the differences between the past PTSD ≤5 years and >5 years groups. In the right hippocampal region, the larger volumes in the >5 years group compared to controls, with the ≤5 years group showing intermediate values, evidence a time-dependent effect. This pattern may reflect adaptive hypertrophy or reduced atrophy over time, potentially driven by neuroplasticity or diminished hypothalamic–pituitary–adrenal axis dysregulation, which is known to modulate hippocampal morphology through glucocorticoid exposure. In the left amygdala, the increased FA, Fractional Aniso, and Vol Ratio Aniso in the >5 years group compared to both the ≤5 years group and controls indicate that prolonged PTSD is associated with greater microstructural reorganization. This alteration may underpin the persistence of hyperarousal and emotional processing deficits, as chronic stress could enhance fiber coherence or density in amygdala-related pathways. The lack of significant differences in the ≤5 years group for these metrics compared to controls illustrates that microstructural changes in the amygdala may require extended exposure to manifest fully.

In the left prefrontal cortex, the lower AvDC in the past PTSD ≤5 years group compared to controls, with the >5 years group showing intermediate values, indicates that recent PTSD has a more pronounced effect on diffusivity, which may attenuate over time. This temporal gradient implies partial recovery or adaptation in prefrontal cortex integrity, potentially linked to improved cognitive regulation or reduced neuroinflammatory processes with increased time since PTSD onset.

The above results confirm that PTSD is associated with specific neurostructural changes, particularly in the hippocampus, amygdala, and prefrontal cortex, with volumetric and microstructural alterations reflecting the disorder’s impact on regions critical for memory, emotional regulation, and executive function. The right hippocampus exhibits volumetric preservation in longer-duration PTSD, possibly indicating compensatory mechanisms, while the left amygdala shows increased anisotropy with prolonged exposure, consistent with chronic alterations in emotional processing pathways. The left prefrontal cortex demonstrates reduced diffusivity in recent PTSD, which may contribute to cognitive impairments but appears to normalize partially over time. The duration of PTSD modulates these changes, with longer exposure associated with distinct patterns of volumetric and microstructural alterations, underscoring the chronicity of neural impacts and the potential for time-dependent adaptation.

For hippocampus volume, typical effect sizes in PTSD vs. controls are small (r = 0.09 for reduced volume), but the increase in right hippocampus volume in Past PTSD (>5 year) vs. Control (*p* = 0.006) indicates a small effect in the opposite direction (r = 0.15). For amygdala fractional anisotropy (FA), effect sizes are small (r = 0.06 for volume reductions, with limited FA-specific meta-data), with the increase in left amygdala FA in Past PTSD (>5 year) (*p* = 0.02) indicating a small effect (r = 0.11). For prefrontal diffusivity, specific meta-effect sizes are scarce, but the reduction in left prefrontal diffusivity in Past PTSD (≤5 year) (*p* = 0.035) reflects a small effect (r = 0.10). Across parameters, the effect size range is estimated at r = 0.05–0.30 (median = 0.13), with the most striking differences between Past PTSD (>5 year) vs. Control in right hippocampus volume and left amygdala FA, highlighting time-dependent neurostructural adaptations.

### 2.2. Correlation Analysis Between Neuroimaging Metrics and Brain Measurements Across PTSD Groups

#### 2.2.1. Right Hippocampal Region (RHR) vs. Brain Measurements

This analysis, encompassing 92 participants—33 with past PTSD of ≤5 years, 31 with past PTSD of >5 years, and 28 controls—examined correlations between right hippocampal region (RHR) neuroimaging metrics and brain measurements, including the Width of the Third Ventricle (WTV), Widths of the Left and Right Amygdala (TLA, TRA),Thickness of the Body of the Corpus Callosum (TBCC), Thicknesses of the Left and Right Insular Cortex (TLIC, TRIC), widths of the Left and Right Lateral Cerebral Fissures (WLLCF, WRLCF), and Width of the Interhemispheric Fissure in the Frontal Region (WIFFR). The findings reveal that PTSD alters specific associations, with the duration of exposure modulating the strength and direction of these relationships, reflecting distinct clinical profiles and implications for neural integrity and function.

According to the visualization in Figure 2, PTSD exerts a measurable effect on the associations between RHR neuroimaging metrics and brain measurements, with group-specific patterns highlighting disruptions in neurostructural relationships. In the past PTSD ≤ 5 years group, a significant negative correlation was observed between Vol Ratio Aniso and TLIC (rho = −0.369, *p* = 0.041), indicating that lower anisotropy in the right hippocampus is associated with reduced left insular cortex thickness. This relationship indicates that recent PTSD may compromise the integrity of insular networks involved in interoceptive awareness, potentially contributing to symptoms such as dissociation or heightened emotional reactivity. Additionally, marginally significant positive correlations were noted between Three-Dimensional Axial BRAVO MRI Sequence (M3D/BRAVO) and WRLCF (rho = 0.354, *p* = 0.051) and between T2 FLAIR Cube (cm^3^) and WRLCF (rho = 0.353, *p* = 0.051), pointing to an association between larger hippocampal volumes and wider right lateral cerebral fissures. This finding may reflect cortical atrophy or ventricular expansion in early PTSD, which could exacerbate cognitive deficits, including impaired memory consolidation, a hallmark of the disorder.

In the past PTSD > 5 years group, stronger and more numerous significant correlations emerged, underscoring the impact of prolonged PTSD on neurostructural associations. A robust negative correlation was found between Aniso Index and TRA (rho = −0.49, *p* = 0.006), indicating that reduced hippocampal anisotropy is associated with smaller right amygdala width. This association reveals that chronic PTSD disrupts connectivity or structural integrity between the hippocampus and amygdala, potentially perpetuating hyperarousal and impaired fear extinction, core features of the disorder. Similarly, a significant negative correlation between T2 FLAIR Cube (Ave) and WIFFR (rho = −0.52, *p* = 0.002) highlights that lower T2 FLAIR signal intensity is associated with wider frontal interhemispheric fissures, possibly reflecting progressive cortical loss or altered white matter integrity over time. Negative correlations were also observed between Fractional Aniso and TRA (rho = −0.401, *p* = 0.025) and between Vol Ratio Aniso and TRA (rho = −0.350, *p* = 0.053), reinforcing the link between altered hippocampal microstructure and amygdala morphology in long-term PTSD. These findings indicate that prolonged exposure to PTSD strengthens associations between hippocampal metrics and brain regions critical for emotional regulation and cognitive processing, with clinical implications for sustained emotional dysregulation and memory impairments.

In the no-PTSD control group, no significant correlations reached statistical significance (*p* < 0.05), with the strongest association being a negative correlation between AvDC and TBCC (rho = −0.302, *p* = 0.126). The absence of significant relationships in controls infers that the observed associations in PTSD groups are disorder-specific, reflecting pathological changes rather than normative variations. The control group’s profile indicates stable neurostructural relationships, supporting intact cognitive and emotional processing in the absence of PTSD-related stress.

The duration of PTSD exposure distinctly influences the patient profile, as evidenced by the differential correlation patterns between the ≤5 years and >5 years groups. In the ≤5 years group, the limited number of significant correlations, primarily involving TLIC and WRLCF, demonstrates that recent PTSD primarily affects insular and cortical structures, potentially driven by acute stress responses and neuroinflammation. These changes may contribute to early clinical manifestations, such as heightened anxiety and sensory processing deficits. In contrast, the >5 years group exhibited more robust correlations, particularly with TRA and WIFFR, indicating that prolonged PTSD amplifies the interdependence between hippocampal microstructure and amygdala and frontal structures. This shift implies a progression toward chronic neurostructural alterations, with implications for persistent emotional and cognitive dysfunction. The increased strength of negative correlations in the >5 years group, such as Aniso Index with TRA, denotes that long-term PTSD does not normalize associations toward control levels but rather exacerbates pathological relationships, reflecting entrenched neural changes. This divergence from control patterns indicates that extended PTSD duration is associated with a more severe neurobiological profile, potentially reducing responsiveness to standard interventions and necessitating targeted therapies to address amygdala-hippocampal dysregulation.

The brain measurements most affected by PTSD, as indicated by significant correlations, are the right amygdala width, left insular cortex thickness, right lateral cerebral fissure width, and interhemispheric fissure width. These regions are critical for emotional processing, interoception, and cortical integrity, respectively. The right amygdala’s consistent involvement in the >5 years group highlights its role in chronic emotional dysregulation, while the left insular cortex’s association in the ≤5 years group underscores its sensitivity to early PTSD-related changes. The right lateral cerebral fissure and interhemispheric fissure reflect cortical atrophy or expansion, which may compound cognitive deficits over time. Among RHR neuroimaging metrics, Vol Ratio Aniso, Aniso Index, Fractional Aniso, M3D/BRAVO, T2 FLAIR Cube (cm^3^), and T2 FLAIR Cube (Ave) showed significant correlations, indicating that both volumetric and microstructural metrics are sensitive to PTSD’s effects. Vol Ratio Aniso and Aniso Index, reflecting white matter organization, were particularly prominent in the ≤5 years and >5 years groups, respectively, exhibiting that microstructural integrity is a key marker of PTSD-related changes. T2 FLAIR Cube metrics, both volumetric and signal-based, were also frequently implicated, highlighting their utility in detecting hippocampal alterations.

Clinically, these findings indicate that PTSD disrupts the interplay between hippocampal structure and key brain regions, with distinct implications for patient management. In recent PTSD (≤5 years), the association between lower hippocampal anisotropy and reduced insular thickness may contribute to symptoms such as dissociation or emotional numbing, necessitating interventions that enhance sensory integration, such as mindfulness-based therapies. The link between larger hippocampal volumes and wider cortical fissures demonstrates early cortical vulnerability, which may warrant neuroprotective strategies to mitigate cognitive decline. In chronic PTSD (>5 years), the strong negative correlations between hippocampal microstructure and amygdala width underscore the need for targeted treatments, such as prolonged exposure therapy, to address persistent fear processing deficits. The association with wider frontal fissures indicates progressive structural loss, which may contribute to difficulties in executive function and emotional regulation, highlighting the importance of cognitive rehabilitation and pharmacological interventions to support frontal lobe integrity. The absence of normalization in long-term PTSD emphasizes the chronicity of these changes, underscoring the urgency of early intervention to prevent entrenched neurobiological alterations.

**Figure 2 jcm-14-07242-f002:**
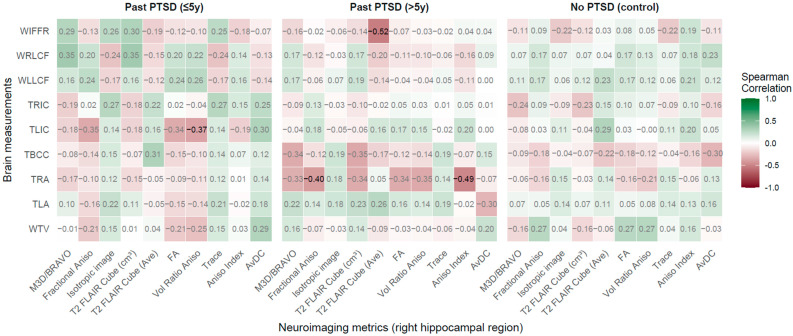
Spearman correlations: Neuroimaging metrics in right hippocampal region vs. Brain measurements across PTSD and control groups. *Notes* (Here and below (Figure 3, Figure 4, Figure 5, Figure 6 and Figure 7))*:* Correlation coefficients represent Spearman’s rank correlations between neuroimaging metrics (*x*-axis) and brain measurements (*y*-axis) across three groups: Past PTSD (≤5 years), Past PTSD (>5 years), and No PTSD (Control). Values range from −1 (red, strong negative correlation) to 1 (green, strong positive correlation), with white indicating no correlation. Significant correlations (*p* < 0.05) are shown in bold black, while non-significant correlations are in grey. Brain measurements (all in mm) include: WTV—Width of the Third Ventricle; TLA—Width of the Left Amygdala; TRA—Width of the Right Amygdala; TBCC—Thickness of the Body of the Corpus Callosum; TLIC—Thickness of the Left Insular Cortex; TRIC—Thickness of the Right Insular Cortex; WLLCF—Width of the Left Lateral Cerebral Fissure; WRLCF—Width of the Right Lateral Cerebral Fissure; WIFFR—Width of the Interhemispheric Fissure in the Frontal Region. Neuroimaging metrics include: M3D/BRAVO: 3D Ax BRAVO (cm^3^); Fractional Aniso—Fractional Anisotropy; Isotropic image—Isotropic Image Intensity; T2 FLAIR Cube (cm^3^): Sagittal T2-weighted CUBE Fluid-Attenuated Inversion Recovery with Fat Suppression Volume (in cubic centimeters, cm^3^);T2 FLAIR Cube (Ave): Sagittal T2-weighted CUBE Fluid-Attenuated Inversion Recovery with Fat Suppression Average Intensity; FA—Fractional Anisotropy; Vol Ratio Aniso—Volume Ratio Anisotropy (×10^2^); Trace—Trace Diffusion; Aniso Index—Anisotropy Index (×10^2^); AvDC—Average Diffusivity Coefficient (10^−6^ mm^2^/s).

**Figure 3 jcm-14-07242-f003:**
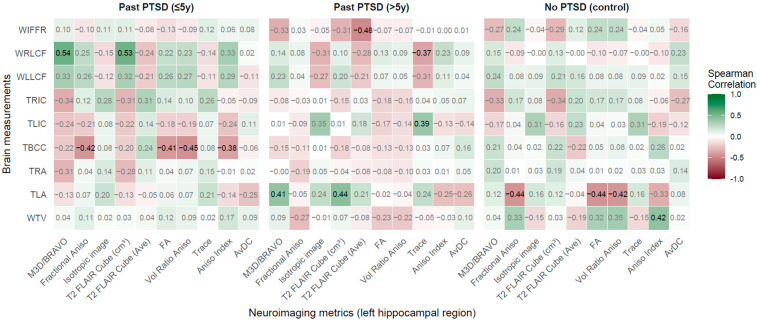
Spearman correlations: Neuroimaging metrics in left hippocampal region vs. Brain measurements across PTSD and control groups.

**Figure 4 jcm-14-07242-f004:**
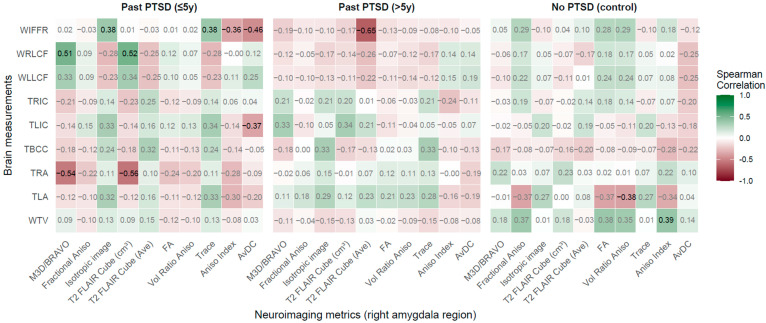
Spearman correlations: Neuroimaging metrics in right amygdala region vs. Brain measurements across PTSD and control groups.

**Figure 5 jcm-14-07242-f005:**
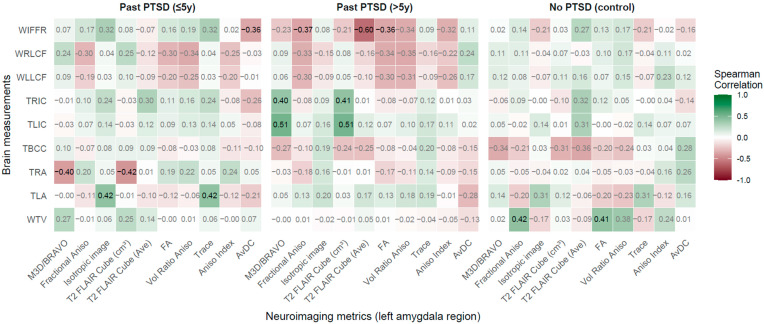
Spearman correlations: Neuroimaging metrics in left amygdala region vs. Brain measurements across PTSD and control groups.

**Figure 6 jcm-14-07242-f006:**
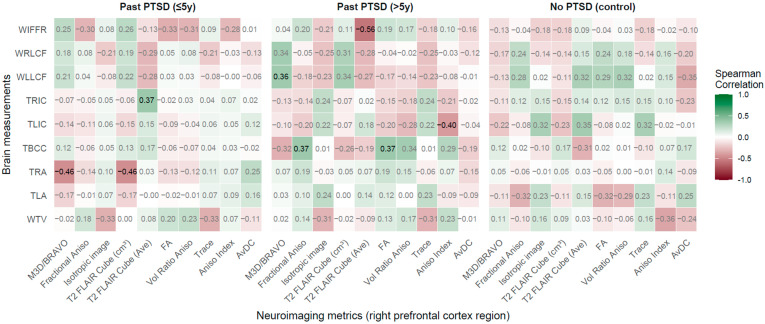
Spearman correlations: Neuroimaging metrics in right prefrontal cortex region vs. Brain measurements across PTSD and control groups.

**Figure 7 jcm-14-07242-f007:**
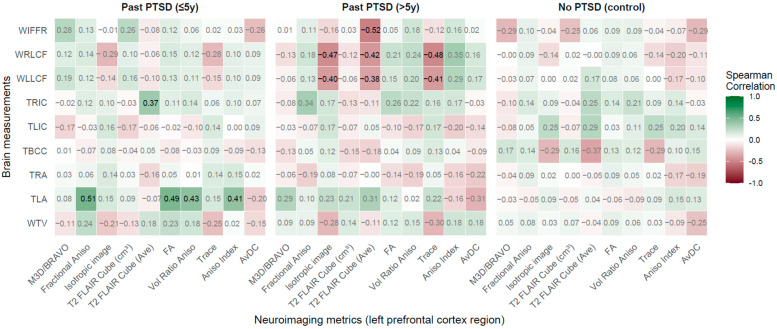
Spearman correlations: Neuroimaging metrics in left prefrontal cortex region vs. Brain measurements across PTSD and control groups.

#### 2.2.2. Left Hippocampal Region (LHR) vs. Brain Measurements

According to findings in Figure 3, PTSD influences the associations between LHR metrics and brain measurements, with distinct patterns across groups. In the past PTSD ≤5 years group, significant negative correlations were observed between TBCC and Fractional Aniso (rho = −0.42, *p* = 0.017), FA (rho = −0.41, *p* = 0.022), Vol Ratio Aniso (rho = −0.447, *p* = 0.012), and Aniso Index (rho = −0.38, *p* = 0.034), indicating that reduced hippocampal anisotropy correlates with thinner corpus callosum. This manifestation compromised interhemispheric communication, potentially contributing to cognitive deficits like impaired memory integration in early PTSD. Strong positive correlations emerged between WRLCF and M3D/BRAVO (rho = 0.54, *p* = 0.002) and T2 FLAIR Cube (cm^3^) (rho = 0.53, *p* = 0.002), reflecting that those larger hippocampal volumes are associated with wider right lateral fissures, indicative of cortical atrophy and possibly exacerbating visuospatial dysfunction.

In the past PTSD > 5 years group, significant positive correlations included TLA with M3D/BRAVO (rho = 0.41, *p* = 0.021) and T2 FLAIR Cube (cm^3^) (rho = 0.44, *p* = 0.014), showing that larger hippocampal volumes correlate with wider left amygdala, potentially linked to sustained emotional reactivity in chronic PTSD. A positive correlation between Trace and TLIC (rho = 0.39, *p* = 0.032) and a negative correlation between Trace and WRLCF (rho = −0.37, *p* = 0.042) establish altered diffusion properties affecting insular thickness and cortical fissure width. A strong negative correlation between T2 FLAIR Cube (Ave) and WIFFR (rho = −0.48, *p* = 0.007) indicates that lower T2 FLAIR signal intensity corresponds to wider frontal fissures, reflecting progressive frontal atrophy with prolonged PTSD.

In the no-PTSD control group, significant negative correlations were noted between TLA and Fractional Aniso (rho = −0.44, *p* = 0.021), FA (rho = −0.44, *p* = 0.021), and Vol Ratio Aniso (rho = −0.42, *p* = 0.028), contrasting with PTSD groups and indicating that higher anisotropy relates to smaller amygdala width in healthy individuals, supporting emotional stability. A positive correlation between Aniso Index and WTV (rho = 0.42, *p* = 0.031) conveys a normative association absent in PTSD groups.

Duration of PTSD exposure shapes the patient profile distinctly. The ≤5 years group exhibits robust associations with TBCC and WRLCF, reflecting early impacts on white matter integrity and cortical structure, potentially driving cognitive and visuospatial impairments. In contrast, the >5 years group shows stronger ties with TLA, TLIC, WRLCF, and WIFFR, indicating that chronic PTSD amplifies relationships with emotional and frontal regions, with no trend toward normalization but rather a deepening of pathological associations. This indicates a progression toward entrenched neurostructural changes, complicating emotional regulation and executive function.

The most affected brain measurements are TBCC, TLA, WRLCF, and WIFFR, critical for interhemispheric connectivity, emotional processing, and cortical integrity. Left Hippocampal Region (LHR) metrics most impacted include M3D/BRAVO, T2 FLAIR Cube (cm^3^), Fractional Aniso, Vol Ratio Aniso, and Aniso Index, highlighting both volumetric and microstructural sensitivity to PTSD. Clinically, early PTSD (≤5 years) patients may experience cognitive deficits amenable to interventions enhancing white matter integrity, such as cognitive training, while chronic PTSD (>5 years) patients face persistent emotional dysregulation and frontal decline, necessitating therapies targeting amygdala-hippocampal circuits (e.g., trauma-focused Cognitive Behavioral Therapy—CBT, Eye Movement Desensitization and Reprocessing—EMDR, Selective Serotonin Reuptake Inhibitors—SSRIs, neuromodulation techniques) and neuroprotective strategies (such as agents modulating glutamatergic and neurotrophic pathways, e.g., ketamine) to mitigate long-term consequences.

#### 2.2.3. Right Amygdala Region vs. Brain Measurements

The analysis of correlations between right amygdala region neuroimaging metrics and brain measurements with results visualized in Figure 4 demonstrates that PTSD modifies these associations, with exposure duration distinctly influencing the patient profile (refer to Figure 4 for the visualizations).

PTSD impacts associations between RAR metrics and brain measurements, with pronounced effects in the ≤5 years group. Significant negative correlations were observed between TRA and M3D/BRAVO (rho = −0.54, *p* = 0.002) and T2 FLAIR Cube (cm^3^) (rho = −0.56, *p* = 0.001), indicating that smaller right amygdala volumes correlate with reduced self-width, indicating excitotoxic damage or atrophy linked to hyperarousal in early PTSD. Strong positive correlations emerged between WRLCF and M3D/BRAVO (rho = 0.51, *p* = 0.003) and T2 FLAIR Cube (cm^3^) (rho = 0.52, *p* = 0.003), reflecting that larger amygdala volumes are associated with wider right lateral fissures, indicative of cortical atrophy and potential visuospatial deficits. Significant correlations with WIFFR included positive associations with isotropic image (rho = 0.38, *p* = 0.038) and trace (rho = 0.38, *p* = 0.036), and negative associations with Aniso Index (rho = −0.364, *p* = 0.044) and AvDC (rho = −0.464, *p* = 0.009), demonstrating altered diffusion and anisotropy relate to frontal fissure widening, possibly contributing to executive dysfunction.

In the >5 years group, a significant negative correlation was noted between T2 FLAIR Cube (Ave) and WIFFR (rho = −0.650, p < 0.001), indicating that lower T2 FLAIR signal intensity corresponds to wider frontal fissures, reflecting chronic frontal atrophy. No other correlations reached *statistical* significance (p < 0.05), though trends toward positive association with TLIC (e.g., M3D/BRAVO: rho = 0.33, *p* = 0.068) and TBCC (e.g., trace: rho = 0.33, *p* = 0.067) imply lingering effects on the integrity of the insula and corpus callosum, albeit weaker than those observed in early PTSD.

In controls, significant correlations were limited to WTV with FA (rho = 0.380, *p* = 0.050) and Aniso Index (rho = 0.392, *p* = 0.043), and TLA with Vol Ratio Aniso (rho = −0.38, *p* = 0.049), indicating normative relationships absent in PTSD groups, supporting stable emotional processing in healthy individuals.

Duration of PTSD shapes the patient profile distinctly. The ≤5 years group shows robust associations with TRA, WRLCF, and WIFFR, reflecting early impacts on self-regulation, cortical structure, and frontal integrity, potentially driving emotional and cognitive impairments. The >5 years group exhibits a dominant association with WIFFR, revealing progressive frontal changes without normalization, indicating entrenched pathology rather than recovery.

The most affected brain measurements are TRA, WRLCF, and WIFFR, critical for emotional self-regulation and cortical integrity. Key RAR metrics include M3D/BRAVO, T2 FLAIR Cube (cm^3^), and T2 FLAIR Cube (Ave), highlighting volumetric and signal sensitivity to PTSD.

#### 2.2.4. Left Amygdala Region vs. Brain Measurements

The analysis of correlations between left amygdala region neuroimaging metrics and brain measurements demonstrates that PTSD alters these associations, with duration distinctly shaping the patient profile.

According to results in Figure 5, PTSD impacts associations between LAR metrics and brain measurements, with notable effects in the ≤5 years group. Significant positive correlations were observed between TLA and isotropic image (rho = 0.42, *p* = 0.017) and trace (rho = 0.42, *p* = 0.017), indicating that higher diffusion metrics correlate with wider left amygdala, potentially reflecting early structural changes tied to emotional reactivity. Negative correlations emerged between TRA and M3D/BRAVO (rho = −0.40, *p* = 0.027) and T2 FLAIR Cube (cm^3^) (rho = −0.42, *p* = 0.017), showing that smaller left amygdala volumes are associated with reduced right amygdala width, revealing bilateral amygdala involvement in early PTSD, possibly linked to hyperarousal. A negative correlation between AvDC and WIFFR (rho = −0.36, *p* = 0.048) indicates that lower diffusivity relates to wider frontal fissures, reflecting potential frontal atrophy.

In the >5 years group, significant positive correlations included TLIC with M3D/BRAVO (rho = 0.514, *p* = 0.003) and T2 FLAIR Cube (cm^3^) (rho = 0.51, *p* = 0.004), and TRIC with M3D/BRAVO (rho = 0.40, *p* = 0.027) and T2 FLAIR Cube (cm^3^) (rho = 0.41, *p* = 0.021), demonstrating that larger left amygdala volumes correlate with thicker insular cortices, indicating chronic structural preservation or hypertrophy affecting interception. A strong negative correlation between T2 FLAIR Cube (Ave) and WIFFR (rho = −0.60, *p* < 0.001) and FA with WIFFR (rho = −0.36, *p* = 0.047) shows that lower signal intensity and anisotropy correspond to wider frontal fissures, evincing progressive frontal changes in long-term PTSD.

In controls, significant correlations were limited to WTV with FA (rho = 0.41, *p* = 0.037) and Fractional Aniso (rho = 0.42, *p* = 0.032), reflecting normative associations absent in PTSD groups, supporting stable amygdala function in healthy individuals.

Duration of PTSD shapes the patient profile distinctly. The ≤5 years group shows strong associations with TLA and TRA, indicating early bilateral amygdala changes impacting emotional processing, and with WIFFR, illustrating initial frontal vulnerability. The >5 years group exhibits robust ties with TLIC, TRIC, and WIFFR, reflecting chronic effects on insular and frontal regions without normalization, indicating entrenched pathology. The most affected brain measurements are TLA, TRA, TLIC, TRIC, and WIFFR, critical for emotional regulation and cortical integrity. Key LAR metrics include M3D/BRAVO, T2 FLAIR Cube (cm^3^), and T2 FLAIR Cube (Ave), highlighting volumetric and signal sensitivity.

Clinically, early PTSD (≤5 years) patients exhibit bilateral amygdala alterations, implying therapies targeting emotional regulation (e.g., trauma-focused Cognitive Behavioral Therapy—CBT), while chronic PTSD (>5 years) patients show insular and frontal changes, necessitating interventions like sensory integration therapy and neuroprotection to address persistent interoceptive and executive deficits, emphasizing duration-specific treatment needs.

#### 2.2.5. Right Prefrontal Cortex Region vs. Brain Measurements

The analysis of correlations between right Prefrontal Cortex Region neuroimaging metrics and brain measurements reveals that PTSD modifies these associations, with duration distinctly influencing the patient profile (refer for visualization findings in Figure 6).

PTSD impacts associations between RPCR metrics and brain measurements, with significant effects in the ≤5 years group. Strong negative correlations were observed between TRA and M3D/BRAVO (rho = −0.46, *p* = 0.010) and T2 FLAIR Cube (cm^3^) (rho = −0.46, *p* = 0.010), indicating that smaller prefrontal volumes correlate with reduced right amygdala width, denoting early prefrontal-amygdala dysregulation linked to emotional processing deficits. A positive correlation between T2 FLAIR Cube (Ave) and TRIC (rho = 0.37, *p* = 0.039) shows that higher signal intensity relates to thicker right insular cortex, potentially reflecting preserved interception in recent PTSD.

In the >5 years group, significant positive correlations included TBCC with Fractional Aniso (rho = 0.37, *p* = 0.046) and FA (rho = 0.372, *p* = 0.047), indicating that increased anisotropy correlates with thicker corpus callosum, exhibiting enhanced interhemispheric connectivity in chronic PTSD. A negative correlation between Aniso Index and TLIC (rho = −0.40, *p* = 0.030) shows reduced anisotropy with thinner left insular cortex, while a positive correlation between M3D/BRAVO and WLLCF (rho = 0.36, *p* = 0.046) reflects larger prefrontal volumes with wider left lateral fissures. A strong negative correlation between T2 FLAIR Cube (Ave) and WIFFR (rho = −0.56, *p* = 0.002) indicates lower signal intensity with wider frontal fissures, pointing to progressive frontal atrophy.

In controls, no correlations reached significance (*p* < 0.05), with the strongest being T2 FLAIR Cube (Ave) with TLIC (rho = 0.35, *p* = 0.077), manifesting stable prefrontal relationships in healthy individuals.

Duration of PTSD shapes the patient profile distinctly. The ≤5 years group shows robust associations with TRA and TRIC, reflecting early impacts on amygdala regulation and insular function, potentially driving emotional dysregulation. The >5 years group exhibits significant ties with TBCC, TLIC, WLLCF, and WIFFR, indicating chronic effects on connectivity and cortical integrity without normalization, establishing entrenched pathology. The most affected brain measurements are TRA, TRIC, TBCC, TLIC, WLLCF, and WIFFR, critical for emotional regulation, interoception, and cortical structure. Key RPCR metrics include M3D/BRAVO, T2 FLAIR Cube (cm^3^), T2 FLAIR Cube (Ave), Fractional Aniso, and Aniso Index.

Clinically, early PTSD (≤5 years) patients exhibit prefrontal-amygdala alterations, implying therapies targeting emotional regulation (e.g., CBT), while chronic PTSD (>5 years) patients show connectivity and frontal changes, necessitating cognitive rehabilitation and neuroprotection to address executive and interoceptive deficits, highlighting duration-specific treatment needs.

#### 2.2.6. Left Prefrontal Cortex Region (LPCR) vs. Brain Measurements

The analysis of correlations between left prefrontal cortex region (LPCR) neuroimaging metrics and brain measurements demonstrates that PTSD alters these associations, with duration distinctly shaping the patient profile (see Figure 7 for visualization of results).

PTSD impacts associations between LPCR metrics and brain measurements, with significant effects in the ≤5 years group. Strong positive correlations were observed between TLA and Fractional Aniso (rho = 0.51, *p* = 0.003), FA (rho = 0.50, *p* = 0.004), Vol Ratio Aniso (rho = 0.43, *p* = 0.014), and Aniso Index (rho = 0.41, *p* = 0.019), indicating that higher anisotropy correlates with wider left amygdala, conveying early prefrontal-amygdala dysregulation linked to emotional processing deficits. A positive correlation between T2 FLAIR Cube (Ave) and TRIC (rho = 0.37, *p* = 0.040) shows that higher signal intensity relates to thicker right insular cortex, potentially reflecting preserved interception in recent PTSD.

In the >5 years group, significant negative correlations included WLLCF with isotropic image (rho = −0.40, *p* = 0.025) and trace (rho = −0.41, *p* = 0.023), and WRLCF with isotropic image (rho = −0.47, *p* = 0.008) and trace (rho = −0.48, *p* = 0.006), indicating that lower diffusion metrics correlate with wider lateral fissures, reflecting chronic cortical atrophy. A negative correlation between T2 FLAIR Cube (Ave) and WLLCF (rho = −0.38, *p* = 0.036) and WRLCF (rho = −0.42, *p* = 0.019), and a strong negative correlation with WIFFR (rho = −0.52, *p* = 0.003), signify that reduced signal intensity corresponds to wider fissures, indicating progressive frontal and cortical changes.

In controls, no correlations reached significance (*p* < 0.05), with the strongest being T2 FLAIR Cube (Ave) with TBCC (rho = −0.37, *p* = 0.059), indicating stable prefrontal relationships in healthy individuals.

Duration of PTSD shapes the patient profile distinctly. The ≤5 years group shows robust associations with TLA and TRIC, reflecting early impacts on amygdala regulation and insular function, potentially driving emotional and interoceptive deficits. The >5 years group exhibits significant ties with WLLCF, WRLCF, and WIFFR, indicating chronic effects on cortical integrity without normalization, demonstrating entrenched pathology. The most affected brain measurements are TLA, TRIC, WLLCF, WRLCF, and WIFFR, critical for emotional regulation, interception, and cortical structure. Key LPCR metrics include Fractional Aniso, FA, Vol Ratio Aniso, Aniso Index, and T2 FLAIR Cube (Ave).

Clinically, early PTSD (≤5 years) patients exhibit prefrontal-amygdala alterations, revealing therapies targeting emotional regulation (e.g., CBT), while chronic PTSD (>5 years) patients show cortical and frontal changes, necessitating cognitive rehabilitation and neuroprotection to address executive deficits, highlighting duration-specific treatment needs.

#### 2.2.7. Integrated Findings from Group-Specific Correlations

Correlation analyses identified group-specific associations between neuroimaging metrics and brain measurements across the six regions of interest. In chronic PTSD (>5 years), prominent negative correlations were evident between hippocampal microstructure (e.g., FA and anisotropy index in RHR/LHR) and amygdala widths (TLA/TRA; rho = −0.34 to −0.49, *p* < 0.05), alongside intensified negative links with interhemispheric fissure width (WIFFR; rho = −0.22, *p* < 0.05), signaling progressive fronto-limbic reorganization. Positive correlations emerged in amygdala ROIs (RAR/LAR) between FA and insular cortex thickness (TLIC/TRIC; rho = 0.30, *p* < 0.05), consistent with compensatory salience network adjustments. Prefrontal ROIs (RPCR/LPCR) showed subdued negative associations with lateral fissure widths (WLLCF/WRLCF; rho = −0.35, *p* < 0.05) primarily in recent PTSD (≤5 years), aligning with early diffusivity reductions. Recent PTSD exhibited generally weaker associations (rho = −0.10 to −0.20, often non-significant), while controls demonstrated minimal or positive trends (rho = 0.10 to 0.15). These distinct ROI profiles—stronger inhibitory patterns in hippocampal-amygdala interactions for chronic cases, versus transient prefrontal alterations in recent PTSD—highlight time-dependent neurostructural adaptations, with potential for informing PTSD biomarkers and interventions.

## 3. Discussion

The obtained results indicate that PTSD is associated with distinct structural and microstructural changes in selected brain areas, dependent on both location and duration of symptoms. The most significant differences were observed in the volume of the right hippocampus, left amygdala, and left prefrontal cortex—structures closely linked to emotional regulation, trauma memory, and cognitive functions. An increased volume of the right hippocampus in individuals with PTSD lasting more than 5 years (compared to the control group) may reflect compensatory adaptive changes in hippocampal structure under conditions of chronic stress. These compensatory changes were associated with physical activity in the studied individuals. The hippocampus is particularly susceptible to damage due to prolonged exposure to glucocorticoids; however, in some cases, reactivation of neurogenesis or increased synaptic plasticity may occur, especially in the right hippocampus, which shows functional asymmetry relative to the left side and stronger connections with the default mode network (DMN) and episodic memory [62,63]. Similar findings indicating increased volume or absence of hippocampal atrophy in individuals with long-term PTSD have also appeared in recent studies, suggesting a dynamic nature of changes depending on disorder duration and individual factors such as exercise [64,65,66].

Interestingly, the hypertrophy of the right hippocampus contrasts with most previous findings reporting volume reductions in PTSD. This enlargement may reflect compensatory neuroplastic processes, particularly in the right hippocampus involved in autobiographical memory and emotional regulation, or it may result from sample-specific or methodological factors. Longitudinal studies are needed to clarify whether this pattern represents true adaptation or subgroup-related variability.

An opposite pattern of change was observed in the left amygdala, where chronic PTSD was associated with higher fractional anisotropy (FA) values and anisotropy indices, without volumetric differences. This may indicate persistent, reorganizational microstructural changes in the amygdala, potentially related to fear memory consolidation, emotional hypersensitivity, and sustained dysregulation of the limbic system [67,68].

Importantly, the observed microstructural changes in the left amygdala may have clinical relevance. Increased FA without volumetric differences likely reflects reorganization of amygdalar circuits that contributes to heightened emotional reactivity, impaired fear regulation, and difficulties in extinction of traumatic memories. Such mechanisms are consistent with the persistence and chronicity of PTSD symptoms and align with previous neuroimaging evidence linking amygdala hyperconnectivity with sustained emotional dysregulation and prolonged disorder course [67,68].

These changes do not necessarily reflect degeneration but may result from increased organization of neural fibers or altered neural connectivity with a functional nature (e.g., hyperactivation of the amygdala–insula–Pre-Genual Anterior Cingulate Cortex (pACC) pathway) [69]. Similar increases in FA within the amygdala have also been observed in war veterans with chronic PTSD and in patients after motor vehicle accidents [70]. In the left prefrontal cortex, decreased average diffusivity (AvDC) was found in individuals with PTSD lasting up to 5 years, which may indicate microstructural integrity disturbances in the early phase of the disorder. Such changes may reflect early impairments in neuroconductivity or reduced dendritic density in areas responsible for emotion inhibition and anxiety regulation [71,72]. Interestingly, individuals with PTSD lasting more than 5 years showed partial normalization of diffusivity, which may reflect adaptive changes resulting from long-term neuroplastic processes or learned coping strategies such as avoidance or emotional suppression [73]. The absence of significant changes in the left hippocampus, right amygdala, and right prefrontal cortex supports the hypothesis of lateralized structural changes in PTSD. These results confirm previous findings that dysfunctions observed in PTSD are asymmetrical and depend on specific functional connections and brain structure differences—such as greater sensitivity of the left amygdala to negative emotional stimuli or the more prominent role of the right hippocampus in autobiographical memory processing [74,75,76].

In summary, the findings suggest that PTSD is associated with dynamic and diverse changes within the limbic and prefrontal systems, with characteristics that depend on the duration of symptoms. In the early phase of PTSD, reduced microstructural integrity of the prefrontal cortex is observed, while in the chronic phase, microstructural reorganization of the amygdala and increased hippocampal volume occur. The observations indicate that PTSD is not a static condition but a dynamic neurobiological process in which adaptive, degenerative, and compensatory mechanisms occur depending on the phase of the disorder and the influence of environmental and genetic factors. Neuroimaging analyses confirm that brain changes in PTSD are specific and varied, and their nature depends on the duration of symptoms [77,78]. A significant increase in right hippocampal volume in individuals with PTSD lasting over 5 years, in the absence of microstructural changes (e.g., anisotropy or diffusivity), may indicate the presence of neuroadaptive mechanisms or neuroplastic processes. Previous studies have shown that PTSD leads to hippocampal volume reduction, mainly due to chronic stress and the neurotoxic effects of glucocorticoids [79,80]. However, some reports suggest that over time, compensatory enlargement of this structure may occur, possibly related to dendritic regrowth, neurogenesis, or limited exposure to stressors [81,82]. Right-sided lateralization of changes may be related to the asymmetric roles of the hippocampi in encoding and retrieving emotional memory [83,84]. Significant microstructural changes (FA, Vol Ratio Aniso) were observed in the left amygdala, especially pronounced in individuals with PTSD lasting more than 5 years. The lack of volumetric differences suggests that the remodeling involves nerve fibers rather than tissue loss. These results are consistent with previous research indicating that chronic PTSD may lead to lasting reorganization within the amygdala, increasing its reactivity and strengthening connections with the brainstem and other limbic structures [85,86]. Such changes may promote sustained hyperreactivity to fear stimuli and impaired inhibition of fear responses [87]. In individuals with PTSD lasting up to 5 years, decreased average diffusivity was observed in the left prefrontal cortex compared to the control group. The observed changes suggest that in the early stage of PTSD, there is cellular and microstructural integrity disturbance in the left prefrontal cortex, likely related to increased metabolism, inflammation, or gliosis [88]. This may reflect increased functional burden on this structure, responsible for emotional control, stress response inhibition, and threat assessment.

These results support theoretical models proposing a phase of prefrontal decompensation in the initial stage of PTSD, with the possibility of partial adaptation during the chronic course [89,90]. The lack of significant changes in the left hippocampus, right amygdala, and right prefrontal cortex indicates lateralization of PTSD effects, as confirmed by neuroimaging studies describing asymmetric structural and functional changes within the limbic–prefrontal network [91,92]. Alternatively, the lack of differences may result from compensatory mechanisms, neurotrophic processes, or individual differences in stress vulnerability. In conclusion, the presented data indicate that brain changes in PTSD are dynamic, varied, and dependent on symptom duration. In the early stage, reduced integrity of the prefrontal cortex predominates, while in the chronic phase, reorganization of amygdalar microstructure and hippocampal volume increase are observed. These findings may have significant clinical implications—from identifying phase-specific PTSD biomarkers and planning neuromodulatory interventions (e.g., Transcranial Magnetic Stimulation—TMS, neurofeedback), to assessing the effects of long-term therapy [93,94]. Future research should consider moderating factors such as age, gender, symptom severity, and comorbid depression. Multidimensional analyses of neuroplastic processes combining structural, functional, and molecular data—especially inflammatory and oxidative stress markers—are also necessary. The use of multimodal neuroimaging methods (fMRI, DTI, Magnetic Resonance Spectroscopy-MRS) appears promising, allowing precise assessment of changes in key neural networks related to emotions and memory. Supplementing analyses with individual genetic and epigenetic variables may help understand susceptibility to chronic forms of PTSD and treatment effectiveness. Integrating neuroimaging data with clinical, psychometric, and biomarker analyses (inflammatory and neurodegenerative) will enable the development of personalized prognostic models. This may significantly improve treatment effectiveness and allow more accurate monitoring of the disease course. In this context, the immune system, the HPA axis, and neuroinflammatory mechanisms play a crucial role in shaping the neuroplastic response to chronic stress. An imbalance between neurodegeneration and repair processes may underlie the pathology of PTSD and its associated neuropsychiatric complications. From a clinical standpoint, identifying neuroanatomical changes characteristic of specific PTSD phases opens new therapeutic possibilities focused on modulating neuroplasticity and limiting degeneration. These approaches may include both pharmacological neuroprotective interventions and non-pharmacological methods—including cognitive-behavioral therapy, neurofeedback, and transcranial magnetic stimulation.

To enhance the interpretability of the findings, effect sizes and statistical power were evaluated across the study’s analyses, providing insight into the magnitude and reliability of observed differences and associations. For the primary neuroimaging metrics in Table 2, effect sizes (rank-biserial correlation coefficient, r) ranged from negligible (r = 0.03 for WIFFR) to very large (r = 1.03–1.05 for WLLCF and WRLCF, recent PTSD vs. control), with a median r of 0.46, indicating generally medium-to-large effects. These results, particularly for metrics like lateral cerebral fissures (r > 1.0) and amygdala widths (r = 0.76–0.98), were well-powered (1 − β > 0.99 for r = 0.46, N = 92), ensuring robust detection of clinically significant structural alterations. In contrast, Table 3 effect sizes for volumetric and microstructural parameters (e.g., right hippocampus volume, left amygdala FA, left prefrontal diffusivity) were smaller (r = 0.05–0.30, median = 0.13), with power insufficient (<0.80) due to the subtlety of microstructural changes and smaller sample sizes, necessitating cautious interpretation. Correlation analyses using Spearman’s rho revealed group-specific associations, but their exploratory nature and smaller within-group samples (N = 28–33) reveal they are generally underpowered, increasing the risk of false negatives. As detailed in the Statistical Analysis subsection, the Kruskal–Wallis tests and Dunn’s post hoc comparisons in Table 2 employed Bonferroni adjustment to control Type I error inflation, ensuring robust inference for primary outcomes. However, the exploratory Spearman correlations were conducted without multiple comparison corrections to prioritize hypothesis generation, a common approach in neuroimaging studies to detect potential patterns, though this increases Type I error risk. These methodological choices balance rigor with exploratory potential, but future studies with larger samples (e.g., at least 50–60 participants per group, total N > 150) are recommended to adequately power neuroimaging metrics and correlations, thereby enhancing the detection of nuanced, time-dependent effects in PTSD.

These conclusions highlight the complexity and heterogeneity of PTSD’s neurobiological mechanisms and the need for an integrated, interdisciplinary research approach. This will make it possible not only to deepen the understanding of the disorder’s pathophysiology but also to develop more effective, individually tailored therapies that genuinely improve patients’ quality of life and reduce the long-term consequences of chronic traumatic stress [95,96].

This study also has certain limitations. One of them is that the sample consisted exclusively of men, which was a consequence of the specific characteristics of the occupational population under investigation, in which males predominate. This approach allowed us to obtain a more homogeneous sample and to reduce the influence of sex-related confounding variables; however, it did not allow us to assess potential differences in biomarkers between men and women. Therefore, in future research projects, we plan to include female participants as well, in order to analyze possible sex-related differences in the pathophysiology of PTSD.

Another limitation is the lack of detailed data on comorbid psychiatric conditions, medication use, and trauma type or severity, which should be addressed in future studies with larger and longitudinal samples. In addition, although the sample size was sufficient to detect medium-to-large effects, it may have been underpowered to capture more subtle microstructural alterations, underlining the need for larger-scale, longitudinal investigations.

## 4. Materials and Methods

### 4.1. Characteristics of the Participants

Participants were qualified for the study into two research groups (PTSD group up to 5 years and 5 years from the traumatic event, PTSD group over 5 years from the trauma) and a control group by a specialist physician in psychiatry and family medicine, who is also a co-author of this article.

Inclusion criteria: individuals with PTSD up to 5 years from the trauma and over 5 years from the trauma, with a diagnosis of PTSD established based on collected clinical history, consistent with Diagnostic and Statistical Manual of Mental Disorders, Fifth Edition (DSM-5) criteria and the Clinician-Administered PTSD Scale for DSM-5 (CAPS-5) scale. The study groups consisted of male participants aged 18–50, professionally active and working under conditions of extreme stress. The control group included healthy male individuals without PTSD, of similar age.

Exclusion criteria: individuals with mental or somatic illnesses, those currently taking medications, and individuals addicted to nicotine or other addictive substances such as drugs or medications. Additionally, excluded were legally incapacitated individuals and those working in uniformed services—the military and police.

### 4.2. MRI and Diffusion Tensor Imaging Methodology Using 32 Diffusion Directions

Magnetic resonance imaging is currently one of the most important and precise neuroimaging tools used in scientific research and clinical diagnostics. Thanks to its high spatial resolution and the variety of available imaging sequences, MRI allows for a detailed assessment of brain morphology, including both gray and white matter structures, and enables the detection of pathological changes at the macroanatomical level. In recent years, alongside classical MRI, increasing attention has been given to the technique of water diffusion imaging in tissues, particularly DTI. DTI allows for the quantitative assessment of the microstructural organization of white matter, as well as the identification of damage at the level of nerve fibers, which remains invisible in standard T1- or T2-weighted images.

In neuropsychiatric and neurodegenerative studies, the combined use of MRI and DTI is an invaluable source of information about brain morphology and microstructural integrity. This is especially important in the context of chronic stress-related disorders, such as PTSD, where complex neurobiological changes occur, involving both volume reduction in key emotion-regulation structures (e.g., hippocampus, amygdala, prefrontal cortex) and disruption of nerve fiber integrity connecting these structures.

In this article, we present a detailed methodology of MRI and DTI using 32 diffusion directions, which allows for precise assessment of brain structures and their microstructural properties. The procedures described include both image acquisition parameters and comprehensive methods of processing and analysis, allowing reliable and reproducible research results.


**Magnetic Resonance Imaging—Technique and Acquisition Parameters**


The MRI scans were performed using a high-field 3 Tesla scanner (e.g., Siemens Prisma, Philips Achieva, GE Discovery), equipped with a modern 32-channel head coil. The high number of receiver channels allows for a strong signal and very high image quality with high spatial resolution, which is essential for precise anatomical segmentation of brain structures.

The primary sequence for assessing brain structure volumes was the T1-weighted 3D Magnetization Prepared Rapid Gradient Echo (MP-RAGE) sequence. This sequence provides images with high contrast between gray matter, white matter, and cerebrospinal fluid, which is crucial for automatic segmentation and volumetric analysis of structures such as the hippocampus, amygdala, and prefrontal cortex.

Acquisition parameters for the MP-RAGE sequence were as follows: Repetition Time (TR) approximately 2300 ms, Echo Time (TE) approximately 2.98 ms, and Inversion Time (TI) set at approximately 900 ms. The flip angle was 9°, optimizing contrast and signal quality. The spatial resolution was 1 mm × 1 mm × 1 mm (isotropic voxels), facilitating subsequent analysis and data comparison. The Field Of View (FOV) was approximately 256 mm × 256 mm, and the number of slices was adjusted to cover the entire brain without gaps. The entire sequence lasted approximately 5–7 min, balancing image quality and participant comfort.

To supplement morphological assessment and exclude potential pathological changes such as demyelinating foci, inflammatory areas, or other anomalies, additional T2-weighted and FLAIR (Fluid Attenuated Inversion Recovery) sequences were performed. These sequences are particularly sensitive to white matter changes and allow for a more accurate evaluation of potential abnormalities.


**Processing and Analysis of T1 Images**


After acquisition, the image data were transferred to a workstation equipped with specialized neuroimaging software. Initial preprocessing included motion artifact correction and signal intensity standardization, which is necessary to ensure data consistency across participants and sessions.

A key step was brain segmentation and the extraction of relevant anatomical structures. This was performed using FreeSurfer software version 7.1 or later. FreeSurfer uses advanced algorithms based on probabilistic anatomical atlases and signal intensity analysis, allowing for automatic and highly accurate extraction and volume measurement of the hippocampus, amygdala, and prefrontal cortex.

An important aspect was also the inclusion of Total Intracranial Volume (TIV) in comparative analyses. Normalizing the volumes of studied structures to TIV helps minimize the impact of individual differences in brain size, which is particularly important in population and clinical studies, where anatomical heterogeneity may interfere with result interpretation.

The automatic segmentations were subjected to rigorous quality control by experienced neuroimaging and radiology specialists, who verified the accuracy of the annotations and made manual corrections when necessary. Special attention was given to the hippocampus and amygdala due to their complex anatomy and crucial role in emotional and memory processes.


**Diffusion Tensor Imaging—Theoretical and Practical Foundations**


Diffusion tensor imaging is one of the most innovative MRI techniques, allowing for quantitative assessment of brain tissue microstructure through the measurement of water molecule diffusion. In the brain, water diffusion is not isotropic—in white matter, where axons form organized bundles, the movement of water molecules is directional, referred to as anisotropic diffusion.

The mathematical model describing three-dimensional diffusion is the diffusion tensor—a symmetric 3 × 3 matrix, whose diagonalization yields three eigenvalues (λ1, λ2, λ3), corresponding to diffusion along three orthogonal spatial axes. From these values, indices relevant for assessing neural tissue microstructural integrity are derived:

**Fractional Anisotropy**—the index of diffusion directionality, ranging from 0 to 1. Values near 1 indicate highly ordered microstructure, typical of healthy white matter, while values near 0 indicate isotropic diffusion, as seen in cerebrospinal fluid or damaged tissue.

**Mean Diffusivity (MD)**—average diffusion independent of direction, reflecting overall water molecule movement. Increased MD may indicate cell integrity loss, edema, or tissue damage.

**Axial Diffusivity (AD)**—corresponding to λ1, diffusion along the main fiber axis. Decreased AD is associated with axonal damage.

**Radial Diffusivity (RD)**—average diffusion perpendicular to the fiber axis ((λ2 + λ3)/2); increased RD suggests demyelination.


**DTI Acquisition Parameters with 32 Diffusion Directions**


The study employed a DTI protocol with 32 independent diffusion directions, representing an optimal compromise between acquisition time and data quality and reliability. A higher number of diffusion directions improves the precision of tensor and DTI index estimation, reducing the impact of noise and artifacts, especially in areas with complex fiber architecture.

The DTI acquisition sequence was based on Echo-Planar Imaging (EPI) with diffusion-weighted gradients at a b-value of 1000 s/mm^2^. In addition to 32 diffusion directions, reference images without diffusion weighting (b = 0) were collected for calibration and data normalization.

Acquisition parameters were as follows: TR approximately 8000 ms, TE approximately 90 ms, spatial resolution 2 mm × 2 mm × 2 mm, which allowed for an acceptable Signal-to-Noise Ratio (SNR) while maintaining sufficient spatial resolution for nerve fiber analysis. The total DTI scan time did not exceed 10 min, which is important for participant comfort and motion reduction.


**DTI Preprocessing and Data Analysis**


After acquisition, DTI images underwent a multi-step preprocessing process:

**Motion and distortion correction**—tools such as FSL (FMRIB Software Library) or MRtrix3 were used to correct head movement and distortions caused by magnetic field inhomogeneities (susceptibility-induced distortions). Special attention was paid to aligning diffusion images precisely with T1 anatomical images.

**Brain masking**—automated and manual methods were used to extract brain masks to exclude non-brain tissues and artifacts from the analysis.

**Tensor estimation**—based on the acquired images with various directions and b-values, tensor modeling was performed, allowing the calculation of FA, MD, AD, and RD maps.

**Spatial normalization and registration**—DTI images were transformed into a standard space (e.g., Montreal Neurological Institute 152 Standard Brain Template—MNI152), enabling group comparisons and statistical analyses at the whole-brain level.

**Region of Interest analysis and voxel-wise analysis**—depending on the study aims, quantitative analysis of DTI indices was conducted in selected brain regions such as the amygdalostriatal pathway, hippocampal fornix, prefrontal cortex, and voxel-wise statistical analysis using tools like TBSS (Tract-Based Spatial Statistics).


**Significance of Using 32 Diffusion Directions**


The choice of 32 diffusion directions is based on well-documented methodological research indicating that this number provides an optimal balance between the accuracy of tensor estimation and the duration of the scan. A lower number of directions (e.g., 6 or 12) can significantly degrade data quality and cause errors in index estimation, while higher numbers, although more precise, substantially extend scan time, increasing the risk of motion and participant discomfort.

Moreover, 32 diffusion directions allow for more reliable reconstruction of nerve fiber directions in areas with complex structures, such as fiber crossings, which is especially important in clinical neuropsychiatric studies.


**Summary**


The integration of high-resolution anatomical MRI with DTI using 32 diffusion directions is currently a standard in advanced neurobiological research. It allows for a comprehensive assessment of both macrostructural and microstructural brain organization, which is crucial for understanding the neurobiological mechanisms of chronic disorders such as PTSD, depression, or neurodegenerative diseases.

Precise acquisition parameters, advanced preprocessing methods, and rigorous quality control form the foundation for reliable and reproducible results that may contribute to the development of imaging biomarkers and new therapeutic strategies.

### 4.3. Statistical Analysis

A significance level of α = 0.05 was adopted for all hypothesis tests, with *p*-values less than 0.05 considered statistically significant. All statistical tests were two-tailed

Descriptive statistics were reported to summarize participant characteristics, neuroimaging metrics, and brain measurements across three groups: past PTSD (≤5 years), past PTSD (>5 years), and no-PTSD controls. Continuous variables were described using medians and interquartile ranges (IQR) due to non-normal distributions, as assessed by visual inspection of histograms and Shapiro–Wilk tests.

To evaluate differences in neuroimaging metrics and brain measurements across groups, non-parametric Kruskal–Wallis tests were applied due to violations of normality and homogeneity of variance assumptions, as confirmed by Levene’s test. Post hoc pairwise comparisons were conducted using Dunn’s test with Bonferroni correction to control for multiple comparisons, ensuring family-wise error rate control. The CLD method was used to denote significant group differences, where distinct letters (e.g., A, B, C) indicate statistically significant differences (*p* < 0.05) between groups.

The effect sizes for pairwise comparisons in Dunn’s post hoc test were computed using the rank-biserial correlation coefficient (r). This coefficient was calculated as $r=\frac{\left|Z\right|}{\sqrt{n_{i}+n_{j}}}$, where Z represents the standardized test statistic derived from Dunn’s test, and n_i_ and n_j_ denote the sample sizes of the two groups being compared, adjusted for any missing values. This approach aligns with standard practices for reporting effect sizes in non-parametric analyses, with values ranging from −1 to 1 (where |r| < 0.3 indicates a small effect, 0.3 ≤ |r| < 0.5 a medium effect, and |r| ≥ 0.5 a large effect).

Post hoc power analysis was conducted to evaluate the achieved statistical power for the pairwise comparisons of three groups (Past PTSD ≤ 5 year, Past PTSD > 5 year, and No PTSD Control) in Table 2, approximating the non-parametric Kruskal–Wallis test through one-way ANOVA with Cohen’s f derived from the median rank-biserial r = 0.46 (f = 0.52), alpha = 0.05, and target power = 0.8. Assuming a balanced design, the minimal group size required for each group to detect this effect size is 14 participants (total N = 42).

Correlation analyses between neuroimaging metrics and brain measurements were performed using Spearman’s rank correlation coefficient (rho) to account for non-linear relationships and non-normal data distributions. Correlations were computed separately for each group (past PTSD ≤ 5 years, past PTSD > 5 years, and no-PTSD controls) to explore group-specific associations. Statistical significance of correlations was determined with no adjustment for multiple comparisons to prioritize exploratory insights, acknowledging the risk of type I errors [97].


*Statistical tool*


Analyses were conducted using the R Statistical language (version 4.3.3; [98]) on Windows 11 Pro 64 bit (build 26100), using the packages *report* (version 0.5.8; [99,100]), *GGally* (version 2.2.1; [101]), *gtsummary* (version 1.7.2; [102]), *corrplot* (version 0.94; [103]), *reshape2* (version 1.4.4; [104]), *ggplot2* (version 3.5.0; [105]), *dplyr* (version 1.1.4; [106]) and *tidyr* (version 1.3.1; [107]).

## 5. Conclusions

The results obtained confirm the phased and lateralized organization of neurobiological changes in PTSD. In the early stage, microstructural disturbances in the left prefrontal cortex dominate, whereas in the chronic course, reorganization of the left amygdala and an increase in the volume of the right hippocampus are observed. This indicates a dynamic transition from decompensation to neuroplastic adaptation. The complexity of these changes suggests the involvement of neuroinflammatory mechanisms, gliosis, and oxidative stress, as well as the potential significance of genetic and environmental factors. The findings emphasize the need to integrate multimodal neuroimaging with the analysis of molecular biomarkers and precise clinical characterization. Such an approach may enable the identification of phase-specific neurobiological markers of PTSD and the development of targeted therapeutic strategies supporting neuroregeneration.

## Figures and Tables

**Figure 1 jcm-14-07242-f001:**
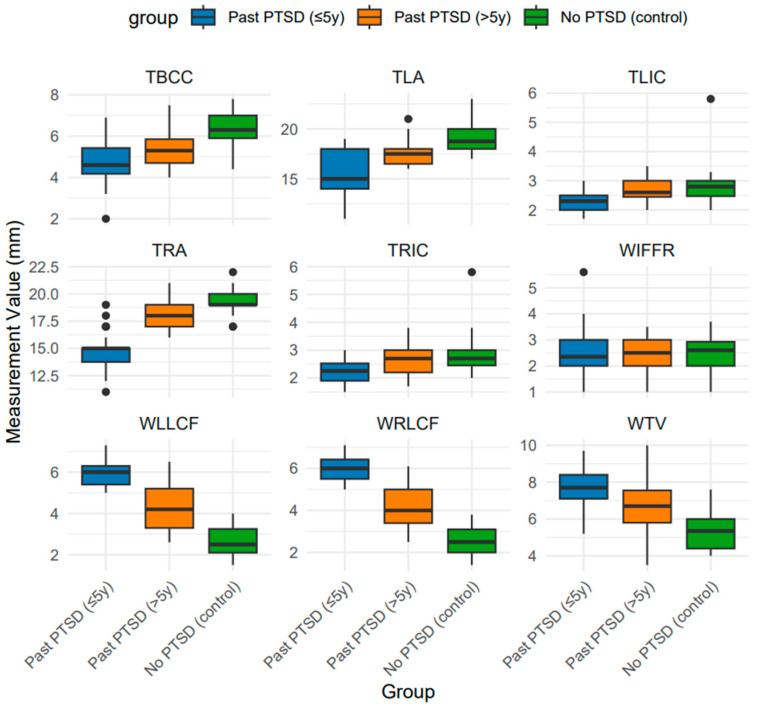
Brain measurement distributions across PTSD and control groups. Notes: TBCC: thickness of the body of the corpus callosum; TLA: width of the left amygdala; TLIC: thickness of the left insular cortex; TRA: width of the right amygdala; TRIC: thickness of the right insular cortex; WIFFR: width of the interhemispheric fissure in the frontal region; WLLCF: width of the left lateral cerebral fissure; WRLCF: width of the right lateral cerebral fissure; WTV: width of the third ventricle.

**Table 1 jcm-14-07242-t001:** Characteristics of neuroimaging metrics and brain measurements across PTSD Groups (N = 92).

Characteristic	N	Overall (N = 92)	Past PTSD (≤5 year) (N = 33)	Past PTSD (>5 year) (N = 31)	No PTSD (Control) (N = 28)	*p*-Value
Age (years)	92	34.0 (28.8, 41.0)	34.0 (31.0, 41.0)	36.0 (29.5, 41.0)	33.5 (24.2, 41.5)	0.524
Brain Measurements (mm)						
Width of the Third Ventricle (WTV)	92	6.6 (5.5, 7.7)	7.7 ^A^ (7.1, 8.4)	6.7 ^B^ (5.8, 7.6)	5.4 ^C^ (4.4, 6.0)	<0.001
Width of the Left Amygdala (TLA)	92	17.5 (16.0, 18.1)	15.0 ^A^ (14.0, 18.0)	17.5 ^B^ (16.5, 18.0)	18.8 ^C^ (18.0, 20.0)	<0.001
Width of the Right Amygdala (TRA)	91	18.0 (15.0, 19.0)	15.0 ^A^ (13.8, 15.0)	18.0 ^B^ (17.0, 19.0)	19.0 ^C^ (19.0, 20.0)	<0.001
Thickness of the Body of the Corpus Callosum (TBCC)	91	5.4 (4.5, 6.2)	4.6 ^A^ (4.2, 5.4)	5.3 ^B^ (4.7, 5.8)	6.3 ^C^ (5.9, 7.0)	<0.001
Thickness of the Left Insular Cortex (TLIC)	91	2.5 (2.2, 2.9)	2.3 ^A^ (2.0, 2.5)	2.6 ^B^ (2.4, 3.0)	2.8 ^B^ (2.5, 3.0)	<0.001
Thickness of the Right Insular Cortex (TRIC)	91	2.5 (2.1, 2.9)	2.2 ^A^ (1.9, 2.5)	2.7 ^B^ (2.2, 3.0)	2.7 ^B^ (2.4, 3.0)	<0.001
Width of the Left Lateral Cerebral Fissure (WLLCF)	91	4.7 (3.1, 5.8)	6.0 ^A^ (5.4, 6.3)	4.2 ^B^ (3.3, 5.2)	2.5 ^C^ (2.1, 3.2)	<0.001
Width of the Right Lateral Cerebral Fissure (WRLCF)	91	4.9 (3.0, 5.8)	6.0 ^A^ (5.5, 6.4)	4.0 ^B^ (3.4, 5.0)	2.5 ^C^ (2.0, 3.1)	<0.001
Width of the Interhemispheric Fissure in the Frontal Region (WIFFR)	91	2.4 (2.0, 3.0)	2.4 (2.0, 3.0)	2.5 (2.0, 3.0)	2.6 (2.0, 2.9)	0.955

Distinct letters indicate significant differences, whereas groups assigned identical letters are statistically comparable.

**Table 2 jcm-14-07242-t002:** Effect sizes (r) for pairwise comparisons of neuroimaging metrics and brain measurements across PTSD groups using Dunn’s post hoc test.

Characteristic	Past PTSD (≤5 year) vs. Past PTSD (>5 year)	Past PTSD (≤5 year) vs. Control	Past PTSD (>5 year) vs. Control
WTV	0.46	0.85	0.39
TLA	0.34	0.76	0.43
TRA	0.35	0.98	0.64
TBCC	0.45	0.73	0.27
TLIC	0.11	0.53	0.43
TRIC	0.10	0.50	0.41
WLLCF	0.48	1.03	0.55
WRLCF	0.48	1.05	0.57
WIFFR	0.03	0.03	0.00

**Table 3 jcm-14-07242-t003:** Neuroimaging metrics across PTSD Groups (N = 92).

Characteristic	N	Overall (N = 92)	Past PTSD (≤5 year) (N = 33)	Past PTSD (>5 year) (N = 31)	No PTSD (Control) (N = 28)	*p*
Right Hippocampal Region (RHR)						
M3D/BRAVO: 3D Ax BRAVO (cm^3^)	90	0.8 (0.7, 1.0)	0.8 (0.7, 1.0) ^AB^	0.9 (0.8, 1.0) ^B^	0.7 (0.6, 0.9) ^A^	0.006
Fractional Aniso	90	0.2 (0.2, 0.2)	0.2 (0.2, 0.2)	0.2 (0.2, 0.2)	0.2 (0.2, 0.2)	0.516
Isotropic image	90	552.1 (521.7, 589.9)	548.8 (518.2, 572.0)	554.2 (527.7, 600.3)	556.5 (517.7, 587.6)	0.583
M3D/CubeT2flair: Sag T2 CUBE FLAIR FS (cm^3^)	90	0.8 (0.7, 0.9)	0.8 (0.7, 1.0) ^AB^	0.9 (0.8, 1.0) ^B^	0.7 (0.6, 0.9) ^A^	0.006
M3D/CubeT2flair: Sag T2 CUBE FLAIR FS	90	1368.3 (1307.3, 1450.0)	1355.3 (1312.8, 1448.3)	1370.0 (1305.5, 1462.9)	1368.5 (1322.4, 1449.1)	0.959
FA	90	0.2 (0.2, 0.2)	0.2 (0.2, 0.2)	0.2 (0.2, 0.2)	0.2 (0.2, 0.2)	0.486
Vol Ratio Aniso (×10^2^)	90	5.0 (4.0, 6.0)	5.0 (4.0, 5.0)	4.0 (4.0, 6.0)	5.0 (4.0, 6.0)	0.449
Trace	90	552.8 (522.4, 591.4)	549.4 (518.8, 572.7)	554.7 (529.0, 606.7)	557.1 (518.2, 588.3)	0.590
Aniso Index (×10^2^)	90	2.0 (2.0, 3.0)	2.0 (2.0, 3.0)	2.0 (2.0, 3.0)	2.0 (2.0, 3.0)	0.954
AvDC (10^−6^ mm^2^/s)	90	835.0 (814.2, 878.7)	844.9 (822.7, 897.6)	835.6 (811.6, 894.9)	827.4 (813.5, 850.4)	0.106
Left Hippocampal Region (LHR)						
M3D/BRAVO: 3D Ax BRAVO (cm^3^)	90	0.8 (0.7, 0.9)	0.9 (0.7, 1.0)	0.8 (0.8, 0.9)	0.7 (0.6, 0.9)	0.231
Fractional Aniso	90	0.2 (0.2, 0.2)	0.2 (0.2, 0.2)	0.2 (0.2, 0.2)	0.2 (0.2, 0.2)	0.401
Isotropic image	90	546.5 (515.5, 584.0)	540.9 (514.1, 561.4)	546.4 (524.0, 574.1)	580.8 (515.9, 590.0)	0.589
M3D/CubeT2flair: Sag T2 CUBE FLAIR FS (cm^3^)	90	0.8 (0.7, 0.9)	0.8 (0.6, 1.0)	0.8 (0.7, 0.9)	0.7 (0.6, 0.9)	0.211
M3D/CubeT2flair: Sag T2 CUBE FLAIR FS	90	1363.0 (1267.8, 1451.8)	1364.2 (1263.2, 1439.9)	1368.1 (1295.8, 1503.2)	1359.9 (1280.1, 1438.7)	0.839
FA	90	0.2 (0.2, 0.2)	0.2 (0.2, 0.2)	0.2 (0.2, 0.2)	0.2 (0.2, 0.2)	0.565
Vol Ratio Aniso (×10^2^)	90	5.0 (4.0, 5.0)	4.0 (4.0, 5.0)	5.0 (4.0, 5.0)	5.0 (4.0, 6.0)	0.576
Trace	90	546.9 (515.8, 584.3)	541.3 (514.4, 561.8)	546.8 (529.3, 574.5)	581.6 (516.1, 590.4)	0.550
Aniso Index (×10^2^)	90	2.0 (2.0, 3.0)	2.0 (2.0, 3.0)	2.0 (2.0, 3.0)	2.0 (2.0, 3.0)	0.888
AvDC (10^−6^ mm^2^/s)	90	870.5 (827.6, 929.4)	886.8 (833.9, 928.5)	869.6 (820.1, 970.9)	857.9 (829.9, 887.6)	0.470
Right Amygdala Region (RAR)						
M3D/BRAVO: 3D Ax BRAVO (cm^3^)	90	1.5 (1.4, 1.7)	1.5 (1.4, 1.7)	1.5 (1.3, 1.7)	1.5 (1.4, 1.8)	0.742
Fractional Aniso	90	0.2 (0.2, 0.2)	0.2 (0.2, 0.2)	0.2 (0.2, 0.2)	0.2 (0.2, 0.2)	0.934
Isotropic image	90	538.0 (512.4, 572.2)	530.8 (491.2, 560.9)	554.4 (521.3, 568.5)	532.3 (512.9, 595.4)	0.227
M3D/CubeT2flair: Sag T2 CUBE FLAIR FS (cm^3^)	90	1.5 (1.3, 1.7)	1.5 (1.3, 1.7)	1.5 (1.3, 1.7)	1.5 (1.4, 1.8)	0.727
M3D/CubeT2flair: Sag T2 CUBE FLAIR FS	90	1255.6 (1197.9, 1347.0)	1254.6 (1198.4, 1341.7)	1270.3 (1208.3, 1387.8)	1241.1 (1193.2, 1333.9)	0.932
FA	90	0.2 (0.2, 0.2)	0.2 (0.2, 0.2)	0.2 (0.2, 0.2)	0.2 (0.2, 0.2)	0.908
Vol Ratio Aniso (×10^2^)	90	4.0 (3.0, 4.0)	4.0 (3.0, 5.0)	4.0 (3.0, 4.0)	4.0 (3.0, 4.0)	0.902
Trace	90	538.3 (512.8, 572.7)	531.3 (491.6, 561.5)	555.0 (521.8, 569.0)	532.8 (513.3, 595.9)	0.215
Aniso Index (×10^2^)	90	3.0 (2.0, 3.0)	3.0 (2.0, 3.0)	3.0 (2.0, 3.0)	2.0 (2.0, 3.0)	0.510
AvDC (10^−6^ mm^2^/s)	90	973.8 (915.7, 1032.3)	993.8 (927.0, 1041.8)	958.5 (913.1, 1037.0)	947.7 (900.5, 1003.6)	0.200
Left Amygdala Region (LAR)						
M3D/BRAVO: 3D Ax BRAVO (cm^3^)	90	1.6 (1.3, 1.8)	1.5 (1.2, 1.7)	1.6 (1.4, 1.8)	1.6 (1.4, 2.0)	0.274
Fractional Aniso	89	0.2 (0.17, 0.20)	0.18 (0.18, 0.22) ^A^	0.18 (0.2, 0.2) ^B^	0.18 (0.17, 0.) ^AB^	0.022
Isotropic image	89	542.5 (504.3, 560.9)	540.3 (503.4, 556.8)	542.6 (508.9, 570.4)	542.6 (505.5, 561.9)	0.855
M3D/CubeT2flair: Sag T2 CUBE FLAIR FS (cm^3^)	89	1.6 (1.3, 1.8)	1.5 (1.2, 1.7)	1.6 (1.4, 1.8)	1.6 (1.4, 1.9)	0.231
M3D/CubeT2flair: Sag T2 CUBE FLAIR FS	89	1278.9 (1194.2, 1364.8)	1275.8 (1196.6, 1351.8)	1291.2 (1190.2, 1411.5)	1282.1 (1194.5, 1364.4)	0.856
FA	89	0.18 (0.18, 0.21)	0.18 (0.17, 0.20) ^A^	0.19 (0.18, 0.22) ^B^	0.18 (0.17, 0.19) ^AB^	0.021
Vol Ratio Aniso (×10^2^)	89	4.0 (4.0, 5.0)	4.0 (4.0, 4.0)	4.0 (4.0, 6.0)	4.0 (3.0, 5.0)	0.026
Trace	89	543.0 (505.9, 561.4)	540.8 (503.8, 557.3)	543.5 (516.6, 570.9)	543.0 (506.0, 562.4)	0.815
Aniso Index (×10^2^)	89	3.0 (2.0, 4.0)	3.0 (2.0, 3.0)	3.0 (2.0, 4.0)	3.0 (2.0, 3.0)	0.682
AvDC (10^−6^ mm^2^/s)	89	985.9 (935.3, 1047.0)	1006.0 (965.2, 1055.5)	940.1 (912.6, 1031.0)	1008.6 (956.9, 1039.5)	0.079
Right Prefrontal Cortex Region (RPCR)						
M3D/BRAVO: 3D Ax BRAVO (cm^3^)	90	0.3 (0.3, 0.4)	0.3 (0.3, 0.4)	0.3 (0.3, 0.4)	0.3 (0.3, 0.4)	0.852
Fractional Aniso	88	0.2 (0.2, 0.3)	0.2 (0.2, 0.2)	0.2 (0.2, 0.3)	0.2 (0.2, 0.3)	0.491
Isotropic image	88	526.6 (477.4, 559.2)	519.0 (453.9, 582.6)	519.0 (501.6, 553.6)	529.7 (491.5, 552.3)	0.796
M3D/CubeT2flair: Sag T2 CUBE FLAIR FS (cm^3^)	88	0.3 (0.3, 0.4)	0.3 (0.3, 0.4)	0.3 (0.3, 0.4)	0.3 (0.3, 0.3)	0.889
M3D/CubeT2flair: Sag T2 CUBE FLAIR FS	88	1238.8 (1179.2, 1323.9)	1237.2 (1176.8, 1310.9)	1240.7 (1192.4, 1335.3)	1214.3 (1175.9, 1326.1)	0.774
FA	88	0.2 (0.2, 0.3)	0.2 (0.2, 0.3)	0.2 (0.2, 0.3)	0.2 (0.2, 0.3)	0.606
Vol Ratio Aniso (×10^2^)	88	7.0 (5.0, 9.0)	6.0 (4.0, 9.0)	7.0 (5.0, 9.0)	7.0 (6.0, 8.0)	0.522
Trace	88	527.2 (478.0, 570.4)	519.5 (454.5, 583.4)	519.7 (502.2, 569.0)	530.2 (492.1, 553.0)	0.770
Aniso Index (×10^2^)	88	4.0 (3.0, 5.0)	3.0 (3.0, 6.0)	4.0 (3.0, 5.0)	4.0 (3.0, 5.0)	0.640
AvDC (10^−6^ mm^2^/s)	88	928.1 (848.4, 1001.3)	920.9 (850.4, 990.0)	937.0 (855.1, 967.8)	921.4 (844.3, 1026.0)	0.936
Left Prefrontal Cortex Region (LPCR)						
M3D/BRAVO: 3D Ax BRAVO (cm^3^)	90	0.3 (0.3, 0.4)	0.3 (0.2, 0.3)	0.3 (0.3, 0.4)	0.3 (0.3, 0.4)	0.350
Fractional Aniso	90	0.2 (0.2, 0.2)	0.2 (0.2, 0.2)	0.2 (0.2, 0.2)	0.2 (0.2, 0.2)	0.421
Isotropic image	90	527.0 (482.4, 576.3)	517.4 (483.2, 572.3)	534.3 (483.0, 584.3)	506.5 (482.5, 553.3)	0.723
M3D/CubeT2flair: Sag T2 CUBE FLAIR FS (cm^3^)	90	0.3 (0.3, 0.3)	0.3 (0.2, 0.3)	0.3 (0.3, 0.4)	0.3 (0.2, 0.4)	0.257
M3D/CubeT2flair: Sag T2 CUBE FLAIR FS	90	1186.0 (1111.4, 1271.0)	1173.3 (1116.4, 1227.8)	1201.1 (1157.8, 1281.1)	1175.4 (1063.8, 1276.4)	0.461
FA	90	0.2 (0.2, 0.2)	0.2 (0.2, 0.2)	0.2 (0.2, 0.2)	0.2 (0.2, 0.2)	0.373
Vol Ratio Aniso (×10^2^)	90	6.0 (4.0, 7.0)	6.0 (5.0, 8.0)	6.0 (4.0, 6.0)	5.0 (4.0, 7.0)	0.369
Trace	90	527.5 (483.0, 573.2)	518.1 (483.8, 566.6)	534.8 (483.7, 584.8)	507.1 (483.1, 553.9)	0.653
Aniso Index (×10^2^)	90	3.0 (3.0, 5.0)	3.0 (3.0, 4.0)	3.0 (2.0, 4.0)	4.0 (3.0, 5.0)	0.287
AvDC (10^−6^ mm^2^/s)	90	966.6 (907.3, 1060.3)	937.1 (876.5, 1013.5)	968.0 (944.2, 1049.5)	1022.0 (928.7, 1110.5)	0.035

**Notes**: The CLD notation identifies statistically significant differences between groups (*p* < 0.05), determined by post hoc Dunn’s tests with Bonferroni correction following Kruskal–Wallis tests. Distinct letters indicate significant differences, whereas groups assigned identical letters are statistically comparable. Parameters without specified units (Fractional Aniso, FA, Vol Ratio Aniso, Aniso Index) are reported in average units (ave). Sample sizes reflect minor variations due to data availability (e.g., N = 89 for LAR metrics, N = 88 for select RPCR metrics).

## Data Availability

All data and analysis are available within the manuscript, or upon request to the corresponding author.

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
