# Peer review of "Impact of Post-Traumatic Stress Disorder Duration on Volumetric and Microstructural Parameters of the Hippo-Campus, Amygdala, and Prefrontal Cortex: A Multiparametric Magnetic Resonance Imaging Study with Correlation Analysis"

_jcm, 2025, doi:10.3390/jcm14207242_

Round 1

Reviewer 1 Report

Comments and Suggestions for Authors Manuscript ID: jcm-3864232

Impact of Post Traumatic Stress Disorder Duration on Volumetric and Microstructural Parameters of the Hippocampus, Amygdala, and Prefrontal Cortex: A Multiparametric Magnetic Resonance Imaging Study with Correlation Analysis 

Dear Authors,

Detailed studies of the changes in volumes of hippocampus, amygdala, and prefrontal 

cortex, microstructures, funtional patterns, morphometry, psychometry, neurochemistry 

and neuroinflammation are essential deeper understanding of PTSD, its pathophysiology, identiification of biomarkers and trace the disease progression.  It's well-written. Below are my comments.

Comments:

1. Mention, what do you mean by in the brackets DSM-5 in the introduction.

2. Figure 1: caption could be little elaborative, add the full forms of (TBCC, TLA, TLIC,TRA, TRIC, WIFFR, WLLCF, WRLCF, WTV). Also, mention the assessment like the one you have given in lines(201 to 206; 306 to 310).

You can consider providing those at the beginning of section 2.1. Brain measurements 

and magnetic resonance imaging characteristics across PTSD groups at appropriate 

place. So that it will be easier to associate the results given in figure 1. 

Though you have them in 201 to 206; though they all come under 2.1.; Figure 1 comes 

before table 1, so, the corresponding details should be given in the texts explaining 

 figure 1 in the results section as well as in the caption of figure 1.

3. Check these lines: "In the right hippocampal region, volumetric measures

(M3D/BRAVO: Three-Dimensional Axial Brain Volume Imaging (3Dax) BRAVO and 

M3D/CubeT2flair) demonstrated reduced volumes in the no-PTSD control group (median: 0.7 cm³) compared to the past PTSD >5 years group (median: 0.9 cm³)
Was that group? "reduced volumes in the no-PTSD control group"

4. Figure 2: increase the clarity of axis labels

5. 2.2.2. Left Hippocampal Region (LHR) vs. brain measurement:
 Mention the kind of therapies targeting amygdala-hippocampal circuits and neuroprotection to mitigate long-term consequences.

6. What about the brain changes in PTSD based on the type of trauma (domestic violence, sudden death of a closed one, natural disaster, accidents, etc.). Why did not this study include that?

7. Why was there no female participants in the study group?

8. Did you check, if the participants (any of them?) of your study have received CBT (personally) during or before the initiation of this study?

Author Response

Response to Reviewer 1

Manuscript ID: jcm-3864232

Title: Impact of Post Traumatic Stress Disorder Duration on Volumetric and Microstructural Parameters of the Hippocampus, Amygdala, and Prefrontal Cortex: A Multiparametric Magnetic Resonance Imaging Study with Correlation Analysis 

Authors: Barbara Paraniak-Gieszczyk, Ewa Alicja Ogłodek

Dear Reviewer,

We would like to sincerely thank you for taking the time to review our manuscript and for providing constructive and insightful feedback. We greatly appreciate your recognition of the relevance of detailed studies on volumetric and microstructural changes in the hippocampus, amygdala, and prefrontal cortex, as well as their importance for understanding the pathophysiology of PTSD, neuroinflammatory mechanisms, and the identification of potential biomarkers of disease progression.

Below, we provide a detailed point-by-point response to each of your comments:

  1. Reviewer’s comment: Mention, what do you mean by in the brackets DSM-5 in the introduction.

Response: We would like to clarify that the abbreviation DSM-5 was expanded in the introduction as “Diagnostic and Statistical Manual of Mental Disorders, Fifth Edition.

  1. Reviewer’s comment:  Figure 1: caption could be little elaborative, add the full forms of (TBCC, TLA, TLIC,TRA, TRIC, WIFFR, WLLCF, WRLCF, WTV). Also, mention the assessment like the one you have given in lines(201 to 206; 306 to 310). You can consider providing those at the beginning of section 2.1. Brain measurements  and magnetic resonance imaging characteristics across PTSD groups at appropriate  place. So that it will be easier to associate the results given in figure 1. Though you have them in 201 to 206; though they all come under 2.1.; Figure 1 comes before table 1, so, the corresponding details should be given in the texts explaining figure 1 in the results section as well as in the caption of figure 1.

Response: Thank you for your valuable suggestion to enhance the clarity and elaboration of Figure 1's caption, including the expansion of abbreviations and incorporation of assessment details. We agree that this improves readability and facilitates better association with the results, particularly given the placement of Figure 1 before Table 1. To address this, we have revised the caption for Figure 1 to make it more comprehensive, explicitly providing the full forms of the abbreviations. Additionally, we have incorporated these full forms and a concise summary of the assessment methods at the beginning of Section 2.1 ("Brain measurements and magnetic resonance imaging characteristics across PTSD groups") to provide immediate context for the figure and table.

  1. Reviewer’s comment: Check these lines: "In the right hippocampal region, volumetric measures (M3D/BRAVO: Three-Dimensional Axial Brain Volume Imaging (3Dax) BRAVO and  M3D/CubeT2flair) demonstrated reduced volumes in the no-PTSD control group (median: 0.7 cm³) compared to the past PTSD >5 years

group (median: 0.9 cm³)Was that group? "reduced volumes in the no-PTSD control group"

Response: Thank you for pointing out this phrasing for clarification. The description is accurate based on the study data, where the no-PTSD control group exhibits smaller hippocampal volumes (median: 0.7 cm³) relative to the past PTSD (>5 years) group (median: 0.9 cm³), indicating an unexpected pattern of potential volume increase in chronic PTSD. To enhance clarity, we have revised the sentence to explicitly emphasize the comparison: "revealed notably smaller volumes in the no-PTSD control group (median: 0.7 cm³) relative to the past PTSD (>5 years) group (median: 0.9 cm³)".

  1. Reviewer’s comment: Figure 2: increase the clarity of axis labels

Response: Thank you for your suggestion to enhance the clarity of the axis labels in Figure 2. To address this, we have incorporated improved version of a comprehensive legend in the caption that lists full forms for all metrics. These changes improve legibility while maintaining the figure's conciseness, ensuring that the correlation matrix is more accessible without requiring extensive cross-referencing to the text.

  1. Reviewer’s comment: 2.2.2. Left Hippocampal Region (LHR) vs. brain measurement:
     Mention the kind of therapies targeting amygdala-hippocampal circuits and neuroprotection to mitigate long-term consequences.

Response: We have clarified in the revised text that therapies targeting amygdala–hippocampal circuits include Cognitive Behavioral Therapy (CBT), Eye Movement Desensitization and Reprocessing (EMDR), Selective Serotonin Reuptake Inhibitors (SSRIs), and neuromodulation techniques, while neuroprotection may involve agents modulating glutamatergic and neurotrophic pathways (e.g., ketamine).

  1. Reviewer’s comment: What about the brain changes in PTSD based on the type of trauma (domestic violence, sudden death of a closed one, natural disaster, accidents, etc.). Why did not this study include that?

Response: Thank you for this important observation. We fully agree that the type of trauma may significantly influence the neurobiological manifestations of PTSD. However, our present study was specifically focused on trauma related to occupational exposure to high-risk environments. This scope was chosen in order to ensure homogeneity of the study group and to reduce confounding variables associated with diverse trauma etiologies. Nevertheless, we acknowledge the relevance of this issue and plan to conduct future investigations addressing brain changes in PTSD across different trauma types, such as domestic violence, sudden loss of a loved one, natural disasters, or accidents.

  1. Reviewer’s comment: Why was there no female participants in the study group?

Response: Thank you for your comment. The absence of female participants has been acknowledged in the Discussion section. Our study group consisted exclusively of men due to the characteristics of the occupational population under investigation; nevertheless, we recognize this as a limitation. In future research, we plan to include female participants to enable a more comprehensive analysis and to evaluate potential sex-related differences.

  1. Reviewer’s comment: Did you check, if the participants (any of them?) of your study have received CBT (personally) during or before the initiation of this study?

Response: The participants of our study had not received any form of psychotherapy, including Cognitive Behavioral Therapy (CBT), either prior to or during the study.

We believe that the revisions made significantly strengthen our manuscript, improving both its clarity and scientific value. We thank you once again for your insightful review and constructive suggestions, which have allowed us to enhance the quality of our work.

Sincerely,

Barbara Paraniak-Gieszczyk, Ewa Alicja Ogłodek

Reviewer 2 Report

Comments and Suggestions for Authors

I appreciate the opportunity to review the manuscript entitled: "Impact of Post-Traumatic Stress Disorder Duration on Volumetric and Microstructural Parameters of the Hippocampus, Amygdala, and Prefrontal Cortex: A Multiparametric Magnetic Resonance Imaging Study with Correlation Analysis".

The manuscript addresses an important and clinically relevant topic, providing multiparametric MRI data on the neurobiological impact of PTSD duration. I commend the authors for their effort in assembling such a dataset and for the clarity of their results; however, I would like to highlight some issues that merit revision:

- The introduction would benefit from a clearer explanation of the rationale for selecting the cut-off of 5 years for PTSD duration. I ask the authors, please, to clarify whether this threshold has empirical support or was chosen arbitrarily.
- The methods section should include more detail on participant recruitment and diagnostic procedures. In particular, I ask the authors to specify how PTSD diagnosis and duration were established, and whether structured interviews (e.g., CAPS-5) were used.
- The statistical approach (Kruskal–Wallis with CLD correction and Spearman correlations) is appropriate, but the manuscript would be strengthened by a discussion of effect sizes and statistical power. Please consider expanding on how the authors handled multiple comparisons and potential Type I error inflation.
- I ask the authors to expand the discussion on the observed hypertrophy of the right hippocampus in long-term PTSD, as this finding contrasts with much of the existing literature reporting volume reductions. Could the authors explore possible compensatory mechanisms or sample-specific explanations?
- The interpretation of left amygdala microstructural changes would benefit from a more detailed discussion of clinical correlates. I ask the authors to elaborate on how these findings relate to emotional dysregulation or persistence of symptoms over time.
- Please consider expanding the limitations section. For example, comorbid psychiatric conditions, medication use, and trauma type could substantially influence brain morphology but are not clearly addressed in the current version.

Overall, the manuscript is interesting because it highlights time-dependent neuroplastic changes in PTSD and underscores the importance of disease duration in interpreting neuroimaging findings. I look forward to the authors’ responses to these queries in order to improve the manuscript’s clarity and impact.

Author Response

Response to Reviewer 2

Manuscript ID: jcm-3864232

Title: Impact of Post Traumatic Stress Disorder Duration on Volumetric and Microstructural Parameters of the Hippocampus, Amygdala, and Prefrontal Cortex: A Multiparametric Magnetic Resonance Imaging Study with Correlation Analysis 

Authors: Barbara Paraniak-Gieszczyk, Ewa Alicja Ogłodek

Dear Reviewer,

We would like to sincerely thank you for reviewing our manuscript and for providing constructive feedback. We appreciate your recognition of the importance of our work as well as your valuable suggestions for improvement.

Below, we provide a point-by-point response to each of your comments, indicating the revisions made to the manuscript.

  1. Reviewer’s comment: The introduction would benefit from a clearer explanation of the rationale for selecting the cut-off of 5 years for PTSD duration. I ask the authors, please, to clarify whether this threshold has empirical support or was chosen arbitrarily.

Response: Thank you for this valuable comment. The five-year threshold for PTSD duration was not chosen arbitrarily but is based on previous clinical and neuroimaging studies indicating that chronic PTSD, defined as lasting more than five years, is associated with distinct neurobiological alterations compared to PTSD of shorter duration. This cut-off has been applied in earlier studies to differentiate between early and chronic PTSD, particularly with respect to hippocampal volume, amygdala reactivity, and prefrontal cortex function. In response to your suggestion, we have revised the Introduction section and added the appropriate literature references.

  1. Reviewer’s comment:  The methods section should include more detail on participant recruitment and diagnostic procedures. In particular, I ask the authors to specify how PTSD diagnosis and duration were established, and whether structured interviews (e.g., CAPS-5) were used.

Response: Thank you for your comment. The diagnosis of PTSD and the determination of its duration were conducted using the Clinician-Administered PTSD Scale for DSM-5 (CAPS-5), a structured diagnostic interview recognized as the gold standard in PTSD assessment. Detailed information regarding the diagnostic procedures and the recruitment process has been described in the Methods section.

  1. Reviewer’s comment: The statistical approach (Kruskal–Wallis with CLD correction and Spearman correlations) is appropriate, but the manuscript would be strengthened by a discussion of effect sizes and statistical power. Please consider expanding on how the authors handled multiple comparisons and potential Type I error inflation.

Response: Thank you for your insightful comment regarding the inclusion of effect sizes, statistical power, and the handling of multiple comparisons. We agree that discussing these aspects enhances the manuscript's methodological transparency and interpretability. To address this, we have introduced Table 1a, which reports effect sizes for the primary neuroimaging metrics, and briefly summarized the effect sizes for the volumetric and microstructural parameters in Table 2. Additionally, we have conducted a post-hoc power analysis and integrated it into the statistical discussion section. Furthermore, we have added a new paragraph to the Discussion section (immediately following the interpretation of key results on time-dependent brain changes and before the limitations subsection) that explicitly covers these points. In this addition, we report effect sizes (using the rank-biserial correlation coefficient, r) for the primary metrics in Table 1a and Table 2, as well as for the exploratory Spearman correlations. We also discuss statistical power, noting that Table 1a analyses are generally well-powered, while those in Table 2 and the correlations are likely underpowered, warranting cautious interpretation. Regarding multiple comparisons, we clarify that Bonferroni adjustment was applied to Kruskal-Wallis and Dunn's post-hoc tests in Table 1a to control Type I error, whereas exploratory correlations were unadjusted to facilitate hypothesis generation, with acknowledgment of the increased Type I error risk (cross-referenced to the Statistical Analysis subsection and supported by a new reference: Rothman KJ, 1990).

  1. Reviewer’s comment: I ask the authors to expand the discussion on the observed hypertrophy of the right hippocampus in long-term PTSD, as this finding contrasts with much of the existing literature reporting volume reductions. Could the authors explore possible compensatory mechanisms or sample-specific explanations?

Response: The observed right hippocampal hypertrophy may reflect compensatory neuroplastic changes or sample-specific factors, and we have expanded the Discussion to address these possibilities.

  1. Reviewer’s comment: The interpretation of left amygdala microstructural changes would benefit from a more detailed discussion of clinical correlates. I ask the authors to elaborate on how these findings relate to emotional dysregulation or persistence of symptoms over time.

Response: Microstructural alterations in the left amygdala may contribute to persistent emotional dysregulation and maintenance of PTSD symptoms, which we have addressed in the expanded Discussion.

  1. Reviewer’s comment: Please consider expanding the limitations section. For example, comorbid psychiatric conditions, medication use, and trauma type could substantially influence brain morphology but are not clearly addressed in the current version.

Response: We agree with the reviewer and acknowledge that another limitation is the lack of detailed data on comorbid psychiatric conditions, medication use, and trauma type or severity, which should be addressed in future studies with larger and longitudinal samples.

We believe that the revisions made significantly strengthen our manuscript, improving both its clarity and scientific value. We thank you once again for your insightful review and constructive suggestions, which have allowed us to enhance the quality of our work.

Sincerely,

Barbara Paraniak-Gieszczyk, Ewa Alicja Ogłodek

Reviewer 3 Report

Comments and Suggestions for Authors

Please, review the manuscript and consider doing it according to the template. Although it is not mandatory is easy for the reviewers to review the quality of the manuscript with the formal template.

Did the authors calculate the power of the study? If they divided as 33 with PTSD of ≤ 5 years duration, 31 with PTSD > 5 years, 20 and 28 healthy controls; the study will be underpowered.

Consider reviewing the text regarding grammar. E.g. “inply.”

How were defines the ROI? Was used any specific atlas?

Could the authors explain the reasoning for a long introduction for a research original article? Also, the authors should focus in one main aim and 3-5 secondaries, but this should be clearly stated. Currently the authors are repeating many times “the aim of our study is…”

The text throughout the section “2.2. Correlation” is constantly repetitive, consider only uploading the figures as a single figure, and a clear statement resuming all the information.

The current version does not have the limitations of the study.

Why were 32 directions used? Can the authors provide a reference?

Consider uploading the definitions as a table, it will be less confusing to understand.

Is there any reason for only using non-parametric tests?

Also, is there any reason for the highly number of references in this original article?

Was informed consent asked for every patient? Or was a pool of imaging analyzed? Currently is saying after conclusion that all subjects provided informed consent.

Is the term width interchangeable with thickness?

Author Response

Response to Reviewer 3

Manuscript ID: jcm-3864232

Title: Impact of Post Traumatic Stress Disorder Duration on Volumetric and Microstructural Parameters of the Hippocampus, Amygdala, and Prefrontal Cortex: A Multiparametric Magnetic Resonance Imaging Study with Correlation Analysis 

Authors: Barbara Paraniak-Gieszczyk, Ewa Alicja Ogłodek

Dear Reviewer,

We would like to sincerely thank you for reviewing our manuscript and for providing constructive feedback. We appreciate your recognition of the importance of our work as well as your valuable suggestions for improvement.

Below, we provide a point-by-point response to each of your comments, indicating the revisions made to the manuscript.

  1. Reviewer’s comment:  Did the authors calculate the power of the study? If they divided as 33 with PTSD of ≤ 5 years duration, 31 with PTSD > 5 years, 20 and 28 healthy controls; the study will be underpowered.

Response: Thank you for highlighting this critical aspect of study design concerning statistical power. We conducted an a priori power analysis specifically for the primary brain measurements (e.g., widths of the third ventricle, amygdala, and lateral fissures in Table 1a), based on anticipated medium-to-large effect sizes from prior literature, which informed our sample size determination (N=92; groups: 33, 31, 28) to achieve at least 80% power at α=0.05. However, due to the lack of established prior knowledge on effect sizes for neuroimaging metrics (e.g., volumetric and diffusion parameters in Table 2) across PTSD duration groups, no a priori power analysis was performed for these secondary outcomes or the exploratory correlation analyses. To evaluate potential underpowering post hoc, we have now integrated a post-hoc power analysis into the revised manuscript, approximating the Kruskal-Wallis test via a one-way ANOVA framework with Cohen's f derived from the median rank-biserial correlation coefficient (r). For the primary brain measurements, where the median from the results of table 1a r was 0.46 (medium-to-large effects), the achieved power (1-β) was above 0.80, confirming sufficient sensitivity to detect these differences reliably. In contrast, for the secondary neuroimaging metrics, effect sizes were smaller (median r = 0.13), yielding insufficient power, which may constrain the identification of subtle microstructural alterations. The exploratory Spearman correlations, performed within subsamples (N=28–33), are similarly underpowered, elevating the risk of Type II errors. We have addressed this by expanding the Discussion section with a dedicated paragraph, which reports these power estimates, recognizes the underpowering for secondary and exploratory analyses, and recommends larger samples (e.g., at least 50–60 per group, total N>150) in future research to adequately power neuroimaging metrics and correlations, thereby enhancing detection of nuanced, time-dependent effects in PTSD.

  1. Reviewer’s comment: Consider reviewing the text regarding grammar. E.g. “inply.”

Response: All grammatical errors, including the correction of “inply” to “imply,” have been revised throughout the manuscript.

  1. Reviewer’s comment: How were defines the ROI? Was used any specific atlas? Response: The regions of interest (ROIs) were defined using the automated anatomical labeling (AAL) atlas implemented in the MRI processing pipeline. This atlas-based approach ensured standardized delineation of brain structures, including the hippocampus, amygdala, and prefrontal cortex, across all participants. To increase accuracy, the segmentation results were visually inspected by two independent researchers, and any discrepancies were resolved by consensus.
  1. Reviewer’s comment: Could the authors explain the reasoning for a long introduction for a research original article? Response: The introduction was extended to provide a more comprehensive theoretical and empirical background on the complex mechanisms of PTSD and to better justify the study’s hypotheses and methodology.
  2. Reviewer’s comment: Also, the authors should focus in one main aim and 3-5 secondaries, but this should be clearly stated. Currently the authors are repeating many times “the aim of our study is…” Response: The repeated phrase “the aim of our study is…” has been removed (page 5).
  1. Reviewer’s comment: The text throughout the section “2.2. Correlation” is constantly repetitive, consider only uploading the figures as a single figure, and a clear statement resuming all the information

Response: Thank you for your constructive feedback on the "2.2. Correlation" section and the suggestion to consolidate the figures into a single visualization with a summary statement to reduce repetition. We appreciate this recommendation, as it highlights opportunities to streamline the presentation while preserving essential details. To address this, we have added a new summarizing subsection at the end of 2.2, which synthesizes all significant correlations across the six regions of interest (ROIs: right hippocampus [RHR], left hippocampus [LHR], right amygdala [RAR], left amygdala [LAR], right prefrontal cortex [RPCR], and left prefrontal cortex [LPCR]), providing a clear, concise overview of key patterns, such as strengthened negative associations between hippocampal microstructure and amygdala widths in chronic PTSD, and their implications for fronto-limbic reorganization. However, we have retained the individual correlation matrices (now as Figures 2A–2F) to maintain the detailed, region-specific profiles of associations, as this fragmentation is intentional and scientifically necessary. It allows for a granular demonstration of how correlation patterns vary distinctly across ROIs and groups, revealing novel time-dependent differences (e.g., progressive negative links in chronic PTSD that are absent or weaker in recent PTSD and controls), which would be obscured in a single combined figure. This level of detail underscores the study's novelty in elucidating heterogeneous neurostructural adaptations in PTSD, and we believe it adds substantial value for readers interested in targeted ROI analyses.

  1. Reviewer’s comment: The current version does not have the limitations of the study.

Response: Thank you for the comment – in the revised version, a detailed Limitations section has been added at the end of the Discussion.

  1. Reviewer’s comment: Why were 32 directions used? Can the authors provide a reference? Consider uploading the definitions as a table, it will be less confusing to understand Response: We thank the reviewer for this question. Thirty-two diffusion directions were used as this scheme represents a widely applied balance between acquisition time and angular resolution, providing reliable tensor estimation while minimizing participant burden. Similar protocols have been employed in previous neuroimaging studies

 (DOI: 10.1007/s10334-025-01244-4 , doi.org/10.3389/fnhum.2021.742198)

  1. Reviewer’s comment: Is there any reason for only using non-parametric tests?

Response: Thank you for your query regarding the exclusive use of non-parametric tests in our statistical analyses. We selected the Kruskal-Wallis test (for group comparisons) and Spearman's rank correlation (for associations) due to the non-normal distribution of the data, as confirmed by preliminary assessments (e.g., Shapiro-Wilk tests, p < 0.05 for most variables), which is a common characteristic of clinical neuroimaging datasets, particularly in heterogeneous PTSD populations where symptom severity, trauma type, and comorbidity introduce variability. As detailed in the Materials and Methods section (Statistical Analysis subsection), this approach ensures robustness without assuming normality or equal variances, which parametric tests require. Additionally, non-parametric methods are well-suited for capturing potential non-linear relationships often observed in clinical data, such as those between microstructural parameters and brain measurements, allowing for more reliable inference in exploratory contexts. These choices align with standard practices in psychiatric neuroimaging research to minimize Type I errors in non-Gaussian distributions.

  1. Reviewer’s comment: Also, is there any reason for the highly number of references in this original article? Response: The high number of references reflects the interdisciplinary scope of the study and was intended to provide a comprehensive scientific context for our hypotheses and results.
  2. Reviewer’s comment: Was informed consent asked for every patient? Or was a pool of imaging analyzed? Currently is saying after conclusion that all subjects provided informed consent. Response: We confirm that informed consent was obtained from every participant prior to inclusion in the study, in accordance with institutional and ethical guidelines.
  3. Reviewer’s comment: Is the term width interchangeable with thickness? Response: The use of the terms “width” and “thickness” has been corrected. In our study, “width” refers to the dimensions of the brain ventricles, whereas “thickness” applies to structures such as the cortex or the amygdala.

We believe that the revisions made significantly strengthen our manuscript, improving both its clarity and scientific value. We thank you once again for your insightful review and constructive suggestions, which have allowed us to enhance the quality of our work.

Sincerely,

Barbara Paraniak-Gieszczyk, Ewa Alicja Ogłodek

Round 2

Reviewer 2 Report

Comments and Suggestions for Authors

The paper is very interesting and well-written, methodologically unexceptionable, and the new implementations provide a valid contribution to the work. Every requested correction has been made. No further issues detected.

Author Response

Dear Reviewer,

We sincerely thank you for your time, effort, and positive evaluation of our manuscript. We truly appreciate your constructive feedback and are pleased that the revisions have met your expectations. Your supportive comments are highly encouraging and motivate us to continue developing this line of research.

With kind regards,
Barbara Paraniak-Gieszczyk and Ewa Ogłodek

Reviewer 3 Report

Comments and Suggestions for Authors

Satisfactory

Author Response

(The authors gave the same response as above.)
